# Learning Manifold Implicitly via Explicit Heat-Kernel Learning

**Yufan Zhou,   Changyou Chen,   Jinhui Xu**
Department of Computer Science and Engineering
State University of New York at Buffalo
{yufanzho, changyou, jinhui}@buffalo.edu

## Abstract

Manifold learning is a fundamental problem in machine learning with numerous applications. Most of the existing methods directly learn the low-dimensional embedding of the data in some high-dimensional space, and usually lack the flexibility of being directly applicable to down-stream applications. In this paper, we propose the concept of *implicit* manifold learning, where manifold information is implicitly obtained by learning the associated heat kernel. A heat kernel is the solution of the corresponding heat equation, which describes how "heat" transfers on the manifold, thus containing ample geometric information of the manifold. We provide both practical algorithm and theoretical analysis of our framework. The learned heat kernel can be applied to various kernel-based machine learning models, including deep generative models (DGM) for data generation and Stein Variational Gradient Descent for Bayesian inference. Extensive experiments show that our framework can achieve state-of-the-art results compared to existing methods for the two tasks.

## 1   Introduction

Manifold is an important concept in machine learning, where a typical assumption is that data are sampled from a low-dimensional manifold embedded in some high-dimensional space. There have been extensive research trying to utilize the hidden geometric information of data samples [1, 2, 3]. For example, Laplacian eigenmap [1], a popular dimensionality reduction algorithm, represents the low-dimensional manifold by a graph built based on the neighborhood information of the data. Each data point serves as a node in the graph, edges are determined by methods like k-nearest neighbors, and weights are computed using Gaussian kernel. With this graph, one can then compute its essential information such as graph Laplacian and eigenvalues, which can help embed the data points to a k-dimensional space (by using the $k$ smallest non-zero eigenvalues and eigenvectors), following the principle of preserving the proximity of data points in both the original and the embedded spaces. Such an approach ensures that as the number of data samples goes to infinity, graph Laplacian converges to the Laplacian-Beltrami operator, a key operator defining the heat equation used in our approach.

In deep learning, there are also some methods try to directly learn the Riemannian metric of manifold other than its embedding. For example, [4] and [5] approximate the Riemannian metric by using the Jacobian matrix of a function, which maps latent variables to data samples.

Different from the aforementioned existing results that try to learn the embedding or Riemannian metric directly, we propose to learn the manifold implicitly by explicitly learning its associated heat kernel. The heat kernel describes the heat diffusion on a manifold and thus encodes a great deal of geometric information of the manifold. Note that unlike Laplacian eigenmap that relies on graph construction, our proposed method targets at a lower-level problem of directly learning the geometry-

encoded heat kernel, which can be subsequently used in Laplacian eigenmap or diffusion map [1, 6], where a kernel function is required. Once the heat kernel is learned, it can be directly applied to a large family of kernel-based machine learning models, thus making the geometric information of the manifold more applicable to down-stream applications. There is a recent work [7] utilizing heat kernel in variational inference, which uses a very different approach from ours. Specifically, our proposed framework approximates the unknown heat kernel by optimizing a deep neural network, based on the theory of Wasserstein Gradient Flows (WGFs) [8]. In summary, our paper has the following main contributions.

- We introduce the concept of implicit manifold learning to learn the geometric information of the manifold through heat kernel, propose a theoretically grounded and practically simple algorithm based on the WGF framework.

- We demonstrate how to apply our framework to different applications. Specifically, we show that DGMs like MMD-GAN [9] are special cases of our proposed framework, thus bringing new insights into Generative Adversarial Networks (GANs). We further show that Stein Variational Gradient Descent (SVGD) [10] can also be improved using our framework.

- Experiments suggest that our proposed framework achieves the state-of-the-art results for applications including image generation and Bayesian neural network regression with SVGD.

**Relation with traditional kernel-based learning**   Our proposed method is also related to kernel selection and kernel learning, and thus can be used to improve many kernel based methods. Compared to pre-defined kernels, our learned kernels can seamlessly integrate the geometric information of the underlying manifold. Compared to some existing kernel-learning methods such as [11, 12], our framework is more theoretically motivated and practically superior. Furthermore, [11, 12] learn kernels by maximizing the Maximum Mean Discrepancy (MMD), which is not suitable when there is only one distribution involved, *e.g.*, learning the parameter manifold in Bayesian Inference.

## 2   Preliminaries

### 2.1   Riemannian Manifold

We use $\mathcal{M}$ to denote manifold, and $dim(\mathcal{M})$ to denote the dimensionality of manifold $\mathcal{M}$. We will only briefly introduce the needed concepts, with formal definitions and details provided in the Appendix. A Riemannian manifold, $(\mathcal{M}, g)$, is a real smooth manifold $\mathcal{M} \subset R^d$ associated with an inner product, defined by a positive definite metric tensor $g$, varying smoothly on the tangent space of $\mathcal{M}$. Given an oriented Riemannian manifold, there exists a Riemannian volume element $\mathrm{d}m$ [13], which can be expressed in local coordinates as: $\mathrm{d}m = \sqrt{|g|}\mathrm{d}x^1 \wedge ... \wedge \mathrm{d}x^d$, where $|g|$ is the absolute value of the determinant of the metric tensor's matrix representation; and $\wedge$ denotes the exterior product of differential forms. The Riemannian volume element allows us to integrate functions on manifolds. Let $f$ be a smooth, compactly supported function on manifold $\mathcal{M}$. The integral of $f$ over $\mathcal{M}$ is defined as $\int_{\mathcal{M}} f\mathrm{d}m$. Now we can define the probability density function (PDF) on manifold [14, 15]. Let $\boldsymbol{\mu}$ be a probability measure on $\mathcal{M} \subset R^d$ such that $\boldsymbol{\mu}(\mathcal{M}) = 1$. A PDF $p$ of $\boldsymbol{\mu}$ on $\mathcal{M}$ is a real, positive and intergrable function satisfying: $\boldsymbol{\mu}(S) = \int_{\mathbf{x} \in S} p(\mathbf{x})\mathrm{d}m, \forall S \subset \mathcal{M}$.

Ricci curvature tensor plays an important role in Riemannian geometry. It describes how a Riemannian manifold differs from an Euclidean space, represented as the volume difference between a narrow conical piece of a small geodesic ball in manifold and that of a standard ball with a same radius in Euclidean space. In this paper, we will focus on Riemannian manifolds with positive Ricci curvatures.

### 2.2   Heat Equation and Heat Kernel

The key ingredient in implicit manifold learning is heat kernel, which encodes extensive geodesic information of the manifold. Intuitively, the heat kernel $k_{\mathcal{M}}(t, \mathbf{x}_0, \mathbf{x})$ describes the process of heat diffusion on the manifold $\mathcal{M}$, given a heat source $\mathbf{x}_0$ and time $t$. Throughout the paper, when $\mathbf{x}_0$ is assumed to be fixed, we will use $k_{\mathcal{M}}^t(\mathbf{x})$ to denote $k_{\mathcal{M}}(t, \mathbf{x}_0, \mathbf{x})$ for notation simplicity. Heat equation and heat kernel are defined as below.

**Definition 1 ([16])** *Let $(\mathcal{M}, g)$ be a connected Riemannian manifold, and $\Delta$ be the Laplace-Beltrami operator on $\mathcal{M}$. The heat kernel $k_{\mathcal{M}}(t, \mathbf{x}_0, \mathbf{x})$ is the minimal positive solution of the heat equation: $\partial k_{\mathcal{M}}(t, \mathbf{x}_0, \mathbf{x})/\partial t = \Delta_{\mathbf{x}} k_{\mathcal{M}}(t, \mathbf{x}_0, \mathbf{x}), \lim_{t \to 0^+} k_{\mathcal{M}}(t, \mathbf{x}_0, \mathbf{x}) = \delta_{\mathbf{x}_0}(\mathbf{x})$.*

Remarkably, heat kernel encodes a massive amount of geometric information of the manifold, and is closely related to the geodesic distance on the manifold.

**Lemma 1 ([16])** *For an arbitrary Riemannian manifold $\mathcal{M}$, $\log k_{\mathcal{M}}(t, \mathbf{x}_0, \mathbf{x}) \sim -d_{\mathcal{M}}^2(\mathbf{x}_0, \mathbf{x})/(4t)$ as $t \to 0$, where $d_{\mathcal{M}}(\mathbf{x}_0, \mathbf{x})$ is the geodesic distance on manifold $\mathcal{M}$ and $\mathbf{x}_0, \mathbf{x} \in \mathcal{M}$.*

This relation indicates that learning a heat kernel also learns the corresponding manifold implicitly. For this reason, we call our method "implicit" manifold learning. It is known that heat kernel is positive definite [17], contains all the information of intrinsic geometry, fully characterizes shapes up to isometry [18]. As a result, heat kernel has been widely used in computer graphics [18, 19].

## 2.3 Wasserstein Gradient Flows

Let $\mathcal{P}(\mathcal{M})$ denote the space of probability measures on $\mathcal{M} \subset R^d$. Assume that $\mathcal{P}(\mathcal{M})$ is endowed with a Riemannian geometry induced by 2-Wasserstein distance, *i.e.*, the distance between two probability measures $\boldsymbol{\mu}_{\mathcal{M}}, \boldsymbol{\nu}_{\mathcal{M}} \in \mathcal{P}(\mathcal{M})$ is defined as: $d_W^2(\boldsymbol{\mu}_{\mathcal{M}}, \boldsymbol{\nu}_{\mathcal{M}}) = \inf_{\pi \in \Gamma(\boldsymbol{\mu}_{\mathcal{M}}, \boldsymbol{\nu}_{\mathcal{M}})} \int_{\mathcal{M} \times \mathcal{M}} \|\boldsymbol{\mu}_{\mathcal{M}} - \boldsymbol{\nu}_{\mathcal{M}}\|^2 d\pi$, where $\Gamma(\boldsymbol{\mu}_{\mathcal{M}}, \boldsymbol{\nu}_{\mathcal{M}})$ is the set of joint distribution on $\mathcal{M} \times \mathcal{M}$ satisfying the condition that its two marginals equal to $\boldsymbol{\mu}_{\mathcal{M}}$ and $\boldsymbol{\nu}_{\mathcal{M}}$, respectively. Let $F : \mathcal{P}(\mathcal{M}) \to R$ be a functional on $\mathcal{P}(\mathcal{M})$, mapping a probability measure to a real value. Wasserstein gradient flows describe the time-evolution of a probability measure $\boldsymbol{\mu}_{\mathcal{M}}^t$, defined by a partial differential equation (PDE): $\partial \boldsymbol{\mu}_{\mathcal{M}}^t/\partial t = -\nabla_{W_2} F(\boldsymbol{\mu}_{\mathcal{M}}^t)$, for $t > 0$, where $\nabla_{W_2} F(\boldsymbol{\mu}_{\mathcal{M}}) \triangleq -\nabla \cdot (\boldsymbol{\mu}_{\mathcal{M}} \nabla(\partial F/\partial \boldsymbol{\mu}_{\mathcal{M}}))$. Importantly, there is a close relation between WGF and the heat equation on manifold.

**Theorem 2 ([20])** *Let $(\mathcal{M}, g)$ be a connected and complete Riemannian manifold with Riemannian volume element $dm$, and $\mathcal{P}_2(\mathcal{M})$ be the Wasserstein space of probability measures on $\mathcal{M}$ equipped with the 2-Wasserstein distance $d_W^2$. Let $(\boldsymbol{\mu}_{\mathcal{M}}^t)_{t>0}$ be a continuous curve in $\mathcal{P}_2(\mathcal{M})$. Then, the followings are equivalent:*

*1. $(\boldsymbol{\mu}_{\mathcal{M}}^t)_{t>0}$ is a trajectory of the gradient flow for the negative entropy $H(\boldsymbol{\mu}_{\mathcal{M}}^t) = \int_{\mathbf{x} \in \mathcal{M}} \log k_{\mathcal{M}}^t(\mathbf{x}) d\boldsymbol{\mu}_{\mathcal{M}}^t \triangleq F(\boldsymbol{\mu}_{\mathcal{M}}^t)$;*

*2. $\boldsymbol{\mu}_{\mathcal{M}}^t$ is given by $\boldsymbol{\mu}_{\mathcal{M}}^t(S) = \int_{\mathbf{x} \in S} k_{\mathcal{M}}^t(\mathbf{x}) dm$ for $t > 0$, where $(k_{\mathcal{M}}^t)_{t>0}$ is a solution to the heat equation $\partial k_{\mathcal{M}}(t, \mathbf{x}_0, \mathbf{x})/\partial t = \Delta_{\mathbf{x}} k_{\mathcal{M}}(t, \mathbf{x}_0, \mathbf{x}), \lim_{t \to 0^+} k_{\mathcal{M}}(t, \mathbf{x}_0, \mathbf{x}) = \delta_{\mathbf{x}_0}(\mathbf{x})$, satisfying: $H(\boldsymbol{\mu}_{\mathcal{M}}^t) < \infty$, and for $\forall 0 < s_0 < s_1$, $\int_{s_0}^{s_1} \int_{\mathbf{x} \in \mathcal{M}} \|\triangle k_{\mathcal{M}}^t(\mathbf{x})\|^2/k_{\mathcal{M}}^t(\mathbf{x}) dm dt < \infty$.*

Different from [20], We use the term negative entropy instead of relative entropy for clearness, because relative entropy is often referred to KL-divergence. From the 2nd bullet of Theorem 2, one can see that $k_{\mathcal{M}}^t$ is the probability density function of $\boldsymbol{\mu}_{\mathcal{M}}^t$ on $\mathcal{M}$ [14, 15].

## 3 The Proposed Framework

Our intuition of learning the heat kernel (thus learning a manifold implicitly) is inspired by Theorem 2, which indicates that one can learn the probability density function (PDF) on the manifold from the corresponding WGF. To this end, we first define the evolving PDF induced by a WGF.

**Definition 2 (Evolving PDF)** *Let $(\mathcal{M}, g)$ be a connected and complete Riemannian manifold with Riemannian volume element $dm$; $(\boldsymbol{\mu}^t)_{t>0}$ be the trajectory of a WGF of negative entropy $H(\boldsymbol{\mu}^t) = \int_{\mathbf{x} \in \mathcal{M}} \log p^t(\mathbf{x}) d\boldsymbol{\mu}^t$ with initial point $\boldsymbol{\mu}^0$. We call the evolving function $p^t$ satisfying $\boldsymbol{\mu}^t(S) = \int_{\mathbf{x} \in S} p^t(\mathbf{x}) dm (\forall t \geq 0)$ the evolving PDF of $\boldsymbol{\mu}^t$ induced by the WGF.*

In the following, we start with some theoretical foundation of heat-kernel learning, which shows that two evolving PDFs induced by the WGF of negative entropy on a given manifold approaches each other at an exponential convergence rate, indicating the learnability of the heat kernel. We then propose an efficient and practical algorithm, where neural network and gradient descent are applied to learn a heat kernel. Finally, we apply our algorithm for Bayesian inference and DGMs as two down-stream applications. All proofs are provided in the Appendix.

### 3.1 Theoretical Foundation of Heat-Kernel Learning

Our goal in this section is to illustrate the feasibility and convergence speed of heat-kernel learning. We start from the following theorem.

**Theorem 3** *With the same setting as in Definition 2, suppose the manifold has positive Ricci curvature, let $p^t$, $q^t$ be two evolving PDFs induced by the WGF of negative entropy with the corresponding probability measures being $\boldsymbol{\mu}^t$ and $\boldsymbol{\nu}^t$, respectively. If $d_W^2(\boldsymbol{\mu}^0, \boldsymbol{\nu}^0) < \infty$, then $p^t(\mathbf{x}) = q^t(\mathbf{x})$ almost everywhere as $t \to \infty$. Furthermore, $\int_{\mathbf{x} \in \mathcal{M}} \|p^t(\mathbf{x}) - q^t(\mathbf{x})\|^2 \mathrm{d}m$ converges to 0 exponentially fast.*

Theorem 3 is a natural extension of Proposition 4.4 in [20], which says that two trajectories of WGF for negative entropy would approach each other. We extend their results from probability measures to evolving PDFs. Thus, if one can learn the trajectory of an evolving PDF $p^t(\mathbf{x})$, it can be used to approximate the true heat kernel $k_{\mathcal{M}}^t$ (which corresponds to $q^t$ in Theorem 3 by Theorem 2).

One potential issue is that if the heat kernel $k_{\mathcal{M}}^t$ itself converges fast enough to 0 in the time limit of $t \to \infty$, the convergence result in Theorem 3 might be uninformative, because one ends up with an almost zero-output function. Fortunately, we can prove that the convergence rate of $\int_{\mathbf{x} \in \mathcal{M}} \|p^t(\mathbf{x}) - k_{\mathcal{M}}^t(\mathbf{x})\|^2 \mathrm{d}m$ is faster than that of $\int_{\mathbf{x} \in \mathcal{M}} \|k_{\mathcal{M}}^t(\mathbf{x})\|^2 \mathrm{d}m$.

**Theorem 4** *Let $(\mathcal{M}, g)$ be a complete Riemannian manifold without boundary or compact Riemannian manifold with convex boundary $\partial \mathcal{M}$. Suppose the manifold has positive Ricci curvature. Then $\int_{\mathbf{x} \in \mathcal{M}} \|k_{\mathcal{M}}^t(\mathbf{x})\|^2 \mathrm{d}m$ converges to 0 at most polynomially as $t \to \infty$.*

In addition, we can also prove a lower bound of the heat kernel $k_{\mathcal{M}}^t$ for the non-asymptotic case of $\forall t < \infty$. This plays an important role when developing our practical algorithm. We will incorporate the lower bound into optimization by Lagrangian multiplier in our algorithm.

**Theorem 5** *Let $(\mathcal{M}, g)$ be a complete Riemannian manifold without boundary or compact Riemannian manifold with convex boundary $\partial \mathcal{M}$. If $\mathcal{M}$ has positive Ricci curvature bounded by $K$ and its dimension $dim(\mathcal{M}) \geq 1$, we have $k_{\mathcal{M}}^t(\mathbf{x}) \geq \dfrac{\Gamma(dim(\mathcal{M})/2 + 1)}{C(\epsilon)(\pi t)^{dim(\mathcal{M})/2}} \exp(\dfrac{\pi^2 - \pi^2 dim(\mathcal{M})}{(4 - \epsilon)Kt})$ for $\forall \mathbf{x}_0, \mathbf{x} \in \mathcal{M}$ and small $\epsilon > 0$, where $C(\epsilon)$ is a constant depending on $\epsilon > 0$ and $d$ such that $C(\epsilon) \to 0$ as $\epsilon \to 0$, and $\Gamma$ is the gamma function.*

Theorem 5 implies that for any finite time $t$, there is a lower bound of the heat kernel, which depends on the time and manifold shape, and is independent of the distance between $\mathbf{x}_0$ and $\mathbf{x}$. In fact, there also exists an upper bound [21], which depends on the geodesic distance between $\mathbf{x}_0$ and $\mathbf{x}$. However, we will show later that the upper bound has little impact in our algorithm, and is thus omitted here.

### 3.2 A Practical Heat-Kernel Learning Algorithm

We now propose a practical framework to solve the heat-kernel learning problem. We decompose the procedure into three steps: 1) constructing a parametric function $p_{\boldsymbol{\phi}}^t$ to approximate the $p^t$ in Theorem 3; 2) bridging $p_{\boldsymbol{\phi}}^t$ and the corresponding $\boldsymbol{\mu}_{\boldsymbol{\phi}}^t$; and 3) updating $\boldsymbol{\mu}_{\boldsymbol{\phi}}^t$ by solving the WGF of negative entropy, leading to an evolving PDF $p_{\boldsymbol{\phi}}^t$. We want to emphasize that by learning to evolve as a WGF, the time $t$ is not an explicit parameter to learn.

**Parameterization of $p_{\boldsymbol{\phi}}^t$** We use a deep neural network to parameterize the PDF. Because $k_{\mathcal{M}}^t(\mathbf{x})$ also depends on $\mathbf{x}_0$, we propose to parameterize $p_{\boldsymbol{\phi}}^t$ as a function with two inputs: $p_{\boldsymbol{\phi}}^t(\mathbf{x}_0, \mathbf{x})$, which is the evolving PDF to approximate the heat kernel (with certain initial conditions). To guarantee the positive semi-definite property of a heat kernel, we utilize some existing parameterizations using deep neural networks [9, 11, 12], where [11] is a special case of [12]. We adopt two ways to construct the kernel. The first way is based on [9], where the parametric kernel is constructed as:

$$p_{\boldsymbol{\phi}}^t(\mathbf{x}_0, \mathbf{x}) = \exp(-\|h_{\boldsymbol{\phi}}^t(\mathbf{x}_0) - h_{\boldsymbol{\phi}}^t(\mathbf{x})\|^2) \tag{1}$$

The second way is based on [12], and we construct a parametric kernel as:

$$p_{\boldsymbol{\phi}}^t(\mathbf{x}_0, \mathbf{x}) = \mathbb{E}\{\cos\{(\boldsymbol{\omega}_{\boldsymbol{\psi}_1}^t)^{\mathsf{T}} [h_{\boldsymbol{\phi}_1}^t(\mathbf{x}_0) - h_{\boldsymbol{\phi}_1}^t(\mathbf{x})]\}\} + \mathbb{E}\{\cos\{(\boldsymbol{\omega}_{\boldsymbol{\psi}_2, \mathbf{x}_0, \mathbf{x}}^t)^{\mathsf{T}} [h_{\boldsymbol{\phi}_1}^t(\mathbf{x}_0) - h_{\boldsymbol{\phi}_1}^t(\mathbf{x})]\}\} \tag{2}$$

**Algorithm 1** Heat Kernel Learning

---

**Input:** samples $\{\mathbf{x}_i\}_{i=1}^n$ on the manifold $\mathcal{M}$, kernel parameterized by (1) or (2), hyper-parameters $\alpha$, $\beta$, $\lambda$, time step $\tau = \alpha/2\beta$.
Initialize function $p_{\boldsymbol{\phi}}^0$, compute corresponding $\boldsymbol{\nu} = \tilde{\boldsymbol{\mu}}_{\boldsymbol{\phi}}^0$ by (3).
**for** $k = 1$ **to** $m$ **do**
    Solve (4), where $\tilde{\boldsymbol{\mu}}_{\boldsymbol{\phi}}^{k\tau}$ is computed by (3). Update $\boldsymbol{\nu} \leftarrow \tilde{\boldsymbol{\mu}}_{\boldsymbol{\phi}}^{k\tau}$, where $\tilde{\boldsymbol{\mu}}_{\boldsymbol{\phi}}^{k\tau}$ is computed by (3).
**end for**

---

where $\boldsymbol{\phi} \triangleq \{\boldsymbol{\phi}_1, \boldsymbol{\psi}_1, \boldsymbol{\psi}_2\}$ in (2), $h_{\boldsymbol{\phi}}^t, h_{\boldsymbol{\phi}_1}^t$ are neural networks, $\boldsymbol{\omega}_{\boldsymbol{\psi}_1}^t, \boldsymbol{\omega}_{\boldsymbol{\psi}_2, \mathbf{x}_0, \mathbf{x}}^t$ are samples from some implicit distribution which are constructed using neural networks. Details of implementing (2) can be found in [12].

A potential issue with these two methods is that they can only approximate functions whose maximum value is 1, *i.e.*, $\max_{\mathbf{x} \in \mathcal{M}} p_{\boldsymbol{\phi}}^t(\mathbf{x}_0, \mathbf{x}) = p_{\boldsymbol{\phi}}^t(\mathbf{x}_0, \mathbf{x}_0) = 1, \forall t > 0$. In practice, this can be satisfied by scaling it with an unknown time-aware term, $a_{\mathcal{M}}^t = \max k_{\mathcal{M}}^t$, as $a_{\mathcal{M}}^t p_{\boldsymbol{\phi}}^t$. Because $a_{\mathcal{M}}^t$ depends only on $t$ and $\mathcal{M}$, it can be seen as a constant for fixed time $t$ and manifold $\mathcal{M}$. As we will show later, the unknown term $a_{\mathcal{M}}^t$ will be cancelled, and thus would not affect our algorithm.

**Bridging $p_{\boldsymbol{\phi}}^t$ and $\boldsymbol{\mu}_{\boldsymbol{\phi}}^t$** We rely on the WGF framework to learn the parametrized PDF. To achieve this, note that from Definition 2, $p_{\boldsymbol{\phi}}^t$ and $\boldsymbol{\mu}_{\boldsymbol{\phi}}^t$ are connected by the Riemannian volume element $dm$. Thus, given $dm$, if one is able to solve $\boldsymbol{\mu}_{\boldsymbol{\phi}}^t$ in the WGF, $p_{\boldsymbol{\phi}}^t$ is also readily solved. However, $dm$ is typically intractable in practice. Furthermore, notice that $p_{\boldsymbol{\phi}}^t(\mathbf{x}_j, \mathbf{x}_i)$ is a function with two inputs. This means for $n$ data samples $\{\mathbf{x}_i\}_{i=1}^n$, there are $n$ evolving PDFs $\{p_{\boldsymbol{\phi}}^t(\mathbf{x}_i, \cdot)\}_{i=1}^n$ and $n$ corresponding trajectories $\{\boldsymbol{\mu}_{\boldsymbol{\phi},i}^t\}_{i=1}^n$, to be solved, which is impractical.

To overcome this challenge, we propose to solving the WGF of $\tilde{\boldsymbol{\mu}}_{\boldsymbol{\phi}}^t \triangleq \sum_{i=1}^n \boldsymbol{\mu}_{\boldsymbol{\phi},i}^t/n$, the averaged probability measure of $\{\boldsymbol{\mu}_{\boldsymbol{\phi},i}^t\}_{i=1}^n$. We approximate the averaged measure by Kernel Density Estimation (KDE) [22]: given samples $\{\mathbf{x}_i\}_{i=1}^n$ on a manifold $\mathcal{M}$ and the parametric function $p_{\boldsymbol{\phi}}^t$, we calculate the unnormalized average $\bar{\boldsymbol{\mu}}_{\boldsymbol{\phi}}^t(\mathbf{x}_i) \approx a_{\mathcal{M}}^t \sum_{j=1}^n p_{\boldsymbol{\phi}}^t(\mathbf{x}_j, \mathbf{x}_i)/n$. Consequently, the normalized average satisfying $\sum_{i=1}^n \tilde{\boldsymbol{\mu}}_{\boldsymbol{\phi}}^t(\mathbf{x}_i) = 1$ is formulated as:

$$\tilde{\boldsymbol{\mu}}_{\boldsymbol{\phi}}^t(\mathbf{x}_i) = \frac{\sum_{j=1}^n p_{\boldsymbol{\phi}}^t(\mathbf{x}_j, \mathbf{x}_i)}{\sum_{i=1}^n \sum_{j=1}^n p_{\boldsymbol{\phi}}^t(\mathbf{x}_j, \mathbf{x}_i)}. \tag{3}$$

We can see that the scalar $a_{\mathcal{M}}^t$ is cancelled, and it will not affect our final algorithm.

**Updating $\boldsymbol{\mu}_{\boldsymbol{\phi}}^t$** Finally, we are left with solving $\tilde{\boldsymbol{\mu}}_{\boldsymbol{\phi}}^t$ in the WGF. We follow the celebrated Jordan-Kinderlehrer-Otto (JKO) scheme [23] to solve $\tilde{\boldsymbol{\mu}}_{\boldsymbol{\phi}}^t$ with the discrete approximation (3). The JKO scheme is rephrased in the following lemma under our setting.

**Lemma 6 ([20])** *Consider probability measures in $\mathcal{P}(\mathcal{M})$. Fix a time step $\tau > 0$ and an initial value $\boldsymbol{\mu}^0$ with finite 2nd moment. Recursively define a sequence $(\boldsymbol{\mu}_\tau^n)_{n \in \mathbb{N}}$ of local minimizer by $\boldsymbol{\mu}_\tau^0 := \boldsymbol{\mu}^0$, $\boldsymbol{\mu}_\tau^n := \arg\min_\eta H(\eta) + d_W^2(\boldsymbol{\mu}_\tau^{n-1}, \eta)/(2\tau)$, where $d_W^2$ denotes the 2-Wasserstein distance. If we further define a discrete trajectory: $\bar{\boldsymbol{\mu}}_\tau^0 := \boldsymbol{\mu}^0$, $\bar{\boldsymbol{\mu}}_\tau^t := \boldsymbol{\mu}_\tau^n$, if $t \in ((n-1)\tau, n\tau]$. Then $\bar{\boldsymbol{\mu}}_\tau^t \to \boldsymbol{\mu}^t$ weakly as $\tau \to 0$ for $\forall t > 0$, where $(\boldsymbol{\mu}^t)_{t>0}$ is a trajectory of the gradient flow of negative entropy $H$.*

Based on Theorem 5 and Lemma 6, we know that to learn the kernel function at time $t < \infty$, we can use the Lagrange multiplier to define the following optimization problem for time $t$:

$$\min_{\boldsymbol{\phi}} \alpha H(\tilde{\boldsymbol{\mu}}_{\boldsymbol{\phi}}^t) + \beta d_W^2(\boldsymbol{\nu}, \tilde{\boldsymbol{\mu}}_{\boldsymbol{\phi}}^t) - \lambda \mathbb{E}_{\mathbf{x}_i \neq \mathbf{x}_j} \left[ p_{\boldsymbol{\phi}}^t(\mathbf{x}_j, \mathbf{x}_i) \right], \tag{4}$$

where $\mathbf{x}_i, \mathbf{x}_j \in \mathcal{M}, \alpha, \beta, \lambda$ are hyper-parameters, time step is $\tau = \alpha/2\beta$, $\boldsymbol{\nu}$ is a given probability measure corresponding to a previous time. The last term is introduced to reflect the constraint of $p_{\boldsymbol{\phi}}^t(\mathbf{x}_j, \mathbf{x}_i)$ reflected in Theorem 5. Also, the Wasserstein term $d_W^2(\boldsymbol{\nu}, \tilde{\boldsymbol{\mu}}_{\boldsymbol{\phi}}^t)$ can be approximated using the Sinkhorn algorithm [24]. Our final algorithm is described in Algorithm 1, some discussions are provided in the Appendix. Note that in practice, mini-batch training is often used to ease computational complexity.

### 3.3 Applications

#### 3.3.1 Learning Kernels in SVGD

SVGD [10] is a particle-based algorithm for approximate Bayesian inference, whose update involves a kernel function $k$. Given a set of particles $\{\mathbf{x}_i\}_{i=1}^n$, at iteration $l$, the particle $\mathbf{x}_i$ is updated by

$$\mathbf{x}_i^{l+1} \leftarrow \mathbf{x}_i^l + \epsilon \phi(\mathbf{x}_i^l), \text{ where } \phi(\mathbf{x}_i^l) = \frac{1}{n} \sum_{j=1}^n \left[ \nabla \log q(\mathbf{x}_j^l) k(\mathbf{x}_j^l, \mathbf{x}_i^l) + \nabla_{\mathbf{x}_j^l} k(\mathbf{x}_j^l, \mathbf{x}_i^l) \right] . \quad (5)$$

Here $q(\cdot)$ is the target distribution to be sampled from. Usually, a pre-defined kernel such as RBF kernel with median trick is used in SVGD. Instead of using pre-defined kernels, we propose to improve SVGD by using our heat-kernel learning method: we learn the evolving PDF and use it as the kernel function in SVGD. By alternating between learning the kernel with Algorithm 1 and updating particles with (5), manifold information can be conveniently encoded into SVGD.

#### 3.3.2 Learning Deep Generative Models

Our second application is to apply our framework to DGMs. Compared to that in SVGD, application in DGMs is more involved because there are actually two manifolds: the manifold of training data $\mathcal{M}_P$ and the manifold of the generated data $\mathcal{M}_{\boldsymbol{\theta}}$. Furthermore, $\mathcal{M}_{\boldsymbol{\theta}}$ depends on model parameters $\boldsymbol{\theta}$, and hence varies during the training process.

Let $g_{\boldsymbol{\theta}}$ denote a generator, which is a neural network parameterized by $\boldsymbol{\theta}$. Let the generated sample be $\mathbf{y} = g_{\boldsymbol{\theta}}(\boldsymbol{\epsilon})$, with $\boldsymbol{\epsilon}$ random noise following some distribution such as the standard normal distribution. In our method, we assume that the learning process constitutes an manifold flow $(\mathcal{M}_{\boldsymbol{\theta}}^s)_{s \geq 0}$ with $s$ representing generator's training step. After each generator update, samples from the generator are assumed to form an manifold. Our goal is to learn a generator such that $\mathcal{M}_{\boldsymbol{\theta}}^\infty$ approaches $\mathcal{M}_P$. Our method contains two steps: learning the generator and learning the kernel (evolving PDF).

**Learning the generator**   We adopt two popular kernel-based quantities as objective functions for our generator, the Maximum Mean Discrepancy (MMD) [25] and the Scaled MMD (SMMD) [26], in which our learned heat kernels are used to compute these quantities. MMD and SMMD can be used to measure the difference between distributions. Thus, we update the generator by minimizing them. Details of the MMD and SMMD are given in the Appendix.

**Learning the kernel**   Different from the simple single manifold setting in Algorithm 1, we consider both the training data manifold and the generated data manifold in learning DGMs. As a result, instead of learning the heat kernel of $\mathcal{M}_{\boldsymbol{\theta}}^s$ or $\mathcal{M}_{\mathcal{P}}$, we propose to learn the heat kernel of a new connected manifold, $\widetilde{\mathcal{M}}^s$, that integrates both $\mathcal{M}_{\boldsymbol{\theta}}^s$ and $\mathcal{M}_{\mathcal{P}}$. We will derive a regularized objective based on (4) to achieve our goal.

The idea is to initialize $\widetilde{\mathcal{M}}^s$ with one of the two manifolds, $\mathcal{M}_{\boldsymbol{\theta}}^s$ and $\mathcal{M}_{\mathcal{P}}$, and then extend it to the other manifold. Without loss of generality, we assume that $\mathcal{M}_{\boldsymbol{\theta}}^s \subseteq \widetilde{\mathcal{M}}^s$ at the beginning. Note that it is unwise to assume $\mathcal{M}_{\boldsymbol{\theta}}^s \cup \mathcal{M}_{\mathcal{P}} \subseteq \widetilde{\mathcal{M}}^s$, since $\mathcal{M}_{\boldsymbol{\theta}}^s$ and $\mathcal{M}_{\mathcal{P}}$ could be very different at the beginning. As a result, $\widetilde{\mathcal{M}}^s = R^d$ might be the only case satisfying $\mathcal{M}_{\boldsymbol{\theta}}^s \cup \mathcal{M}_{\mathcal{P}} \subseteq \widetilde{\mathcal{M}}^s$, which does not contain any useful geometric information. First of all, we start with $\mathcal{M}_{\boldsymbol{\theta}}^s$ by consider $p_{\boldsymbol{\phi}}^t(\mathbf{y}_i, \mathbf{y}_j), \mathbf{y}_i, \mathbf{y}_j \in \mathcal{M}_{\boldsymbol{\theta}}^s \subset \widetilde{\mathcal{M}}^s$ in (4). Next, to incorporate the information of $\mathcal{M}_{\mathcal{P}}$, we consider $p_{\boldsymbol{\phi}}^t(\mathbf{y}, \mathbf{x})$ in (4) and regularize it with $\|p_{\boldsymbol{\phi}}^t(\mathbf{y}, \mathbf{x}) - p_{\boldsymbol{\phi}}^t(\mathbf{y}, \mathbf{z})\|$, where $\mathbf{y} \in \mathcal{M}_{\boldsymbol{\theta}}^s$, $\mathbf{x} \in \mathcal{M}_{\mathcal{P}}$ and $\mathbf{z} \in \widetilde{\mathcal{M}}^s$ is the closest point to $\mathbf{x}$ on $\widetilde{\mathcal{M}}^s$. The regularization constrains $\mathcal{M}_{\mathcal{P}}$ to be closed to $\widetilde{\mathcal{M}}^s$ (extending $\widetilde{\mathcal{M}}^s$ to $\mathcal{M}_{\mathcal{P}}$). Since the norm regularization is infeasible to calculate, we will derive an upper bound below and use that instead. Specifically, for kernels of form (1), by Taylor expansion, we have:

$$\|p_{\boldsymbol{\phi}}^t(\mathbf{y}, \mathbf{x}) - p_{\boldsymbol{\phi}}^t(\mathbf{y}, \mathbf{z})\| \approx \|(\partial p_{\boldsymbol{\phi}}^t(\mathbf{y}, \mathbf{x})/\partial \mathbf{x})(\mathbf{z} - \mathbf{x})\| \leq c p_{\boldsymbol{\phi}}^t(\mathbf{y}, \mathbf{x}) \|\nabla_{\mathbf{x}} h_{\boldsymbol{\phi}}^t(\mathbf{x})\|_{\mathcal{F}} \|h_{\boldsymbol{\phi}}^t(\mathbf{x}) - h_{\boldsymbol{\phi}}^t(\mathbf{y})\| \quad (6)$$

where $c = \|\mathbf{x} - \mathbf{z}\|$, and $\|\cdot\|_{\mathcal{F}}$ denotes the Frobenius norm. Consider $p_{\boldsymbol{\phi}}^t(\mathbf{y}, \mathbf{x})$ in (4) will lead to the same bound because of symmetry.

Finally, we consider $p_{\boldsymbol{\phi}}^t(\mathbf{x}_i, \mathbf{x}_j)$ in (4) and regularize $\|p_{\boldsymbol{\phi}}^t(\mathbf{x}_j, \mathbf{x}_i) - p_{\boldsymbol{\phi}}^t(\mathbf{z}_j, \mathbf{z}_i)\|$, where $\mathbf{x}_i, \mathbf{x}_j \in \mathcal{M}_{\mathcal{P}}$, and $\mathbf{z}_i, \mathbf{z}_j \in \widetilde{\mathcal{M}}^s$ are the closest points to $\mathbf{x}_i$ and $\mathbf{x}_j$ on $\widetilde{\mathcal{M}}^s$. A similar bound can be obtained.

Furthermore, instead of directly bounding the multiplicative terms in (6), we find it more stable to bound every component separately. Note that $\|\nabla_{\mathbf{x}} h_{\boldsymbol{\phi}}^t(\mathbf{x})\|_{\mathcal{F}}$ can be bounded from above using spectral normalization [27] or being incorporated into the objective function as in [26]. We do not explicitly include it in our objective function. As a result, incorporating the base learning kernel from (1), our optimization problem becomes:

$$\min_{\boldsymbol{\phi}} \alpha H(\tilde{\boldsymbol{\mu}}_{\boldsymbol{\phi}}^t) + \beta d_W^2(\boldsymbol{\nu}, \tilde{\boldsymbol{\mu}}_{\boldsymbol{\phi}}^t) - \lambda \mathbb{E}_{\mathbf{y} \neq \mathbf{x}} \left[ p_{\boldsymbol{\phi}}^t(\mathbf{y}, \mathbf{x}) \right] + \mathbb{E}_{\mathbf{x} \sim \mathbb{P}, \mathbf{y} \sim \mathbb{Q}} \left[ \gamma_1 p_{\boldsymbol{\phi}}^t(\mathbf{y}, \mathbf{x}) + \gamma_2 \| h_{\boldsymbol{\phi}}^t(\mathbf{x}) - h_{\boldsymbol{\phi}}^t(\mathbf{y}) \| \right]$$

$$+ \mathbb{E}_{\mathbf{x}_i, \mathbf{x}_j \sim \mathbb{P}} \left[ \gamma_3 p_{\boldsymbol{\phi}}^t(\mathbf{x}_j, \mathbf{x}_i) + \gamma_4 \| h_{\boldsymbol{\phi}}^t(\mathbf{x}_i) - h_{\boldsymbol{\phi}}^t(\mathbf{x}_j) \| \right], \text{ where} \quad (7)$$

$$\mathbb{E}_{\mathbf{y} \neq \mathbf{x}} \left[ p_{\boldsymbol{\phi}}^t(\mathbf{y}, \mathbf{x}) \right] = \frac{1}{4} \{ \mathbb{E}_{\mathbf{y}_i, \mathbf{y}_j \sim \mathbb{Q}} \left[ p_{\boldsymbol{\phi}}^t(\mathbf{y}_j, \mathbf{y}_i) \right] + 2 \mathbb{E}_{\mathbf{x} \sim \mathbb{P}, \mathbf{y} \sim \mathbb{Q}} \left[ p_{\boldsymbol{\phi}}^t(\mathbf{y}, \mathbf{x}) \right] + \mathbb{E}_{\mathbf{x}_i, \mathbf{x}_j \sim \mathbb{P}} \left[ p_{\boldsymbol{\phi}}^t(\mathbf{x}_j, \mathbf{x}_i) \right] \} \quad (8)$$

Our algorithm for DGM with heat kernel learning is shown in Appendix. When SMMD is used as the objective function for the generator, we also scale (7) by the same factor as scaling SMMD.

**Theoretical property and relation with existing methods:** Following the work in [26], we first study the continuity in weak-topology of our kernel when applied in MMD. The continuity in weak topology is an important property because it means that the objective function can provide good signal to update the generator [26], without suffering from sudden jump as in the Jensen-Shannon (JS) divergence or Kullback-Leibler (KL) divergence [28].

**Theorem 7** *With* (6) *bounded, MMD of our proposed kernel is continuous in weak topology, i.e., if* $\mathbb{Q} \xrightarrow{D} \mathbb{P}$ *then* $\mathrm{MMD}_{k_{\boldsymbol{\phi}}^t}(\mathbb{Q}_n, \mathbb{P}) \to 0$*, where* $\xrightarrow{D}$ *means convergence in distribution.*

Proof of Theorem 7 directly follows Theorem 2 in [26]. Plugging (8) into (7), it is interesting to see some connections of existing methods with ours: 1) If one sets $\alpha = \beta = \gamma_2 = \gamma_3 = \gamma_4 = 0, \gamma_1 = 4, \lambda = 4$, our method reduces to MMD-GAN [9]. Furthermore, if the scaled objectives are used, our method reduces to SMMD-GAN [26]; 2) If one sets $\alpha = \beta = \gamma_2 = \gamma_4 = 0, \gamma_1 + \gamma_3 = 4, \lambda = 4$, our method reduces to the MMD-GAN with repulsive loss [29].

In summary, our method can interpret the min-max game in MMD-based GANs from a kernel-learning point of view, where the discriminators try to learn the heat kernel of some underlying manifolds. As we will show in the experiments, our model achieves the best performance compared to the related methods. Although there are several hyper-parameters in (7), we have made our model easy to tune due to the connection with GANs. Specifically, one can start by selecting a kernel based GAN, *e.g.*, setting $\alpha = \beta = \gamma_2 = \gamma_3 = \gamma_4 = 0, \gamma_1 = 4, \lambda = 4$ as MMD-GAN, and only tune $\alpha, \beta$.

## 4 Experiments

### 4.1 A Toy Experiment

We illustrate the effectiveness of our method by comparing the difference between a learned PDF and the true heat kernel on the real line $R$, *i.e.*, the 1-dimensional Euclidean space. In this setting, the heat kernel has a closed form of $k(t, x_0, x) = \exp\{-(x - x_0)^2/4t\}/\sqrt{4\pi t}$, where the maximum value is $a_{\mathcal{M}}^t = 1/\sqrt{4\pi t}$. We uniformly sample 512 points in $[-10, 10]$ as training data, the kernel is constructed by (1), where a 3-layer neural network is used. We assume that every gradient descent update corresponds to 0.01 time step. The evolution of $a_{\mathcal{M}}^t p_{\boldsymbol{\phi}}^t(0, x)$ and $k_{\mathcal{M}}^t$ are shown in Figure 1.

### 4.2 Improved SVGD

We next apply SVGD with the kernel learned by our framework for BNN regression on UCI datasets. For all experiments, a 2-layer BNN with 50 hidden units, 10 weight particles, ReLU activation is used. We assign the isotropic Gaussian prior to the network weights. Recently, [30] proposes the matrix-valued kernel for SVGD (denoted as MSVGD-a and MSVGD-m). Our method can also be used to improve their methods. Detailed experimental settings are provided in the Appendix due to the limitation of space. We denote our improved SVGD as HK-SVGD, and our improved matrix-valued SVGD as HK-MSVGD-a and HK-MSVGD-m. The results are reported in Table 1. We can see that our method can improve both the SVGD and matrix-valued SVGD. Additional test log-likelihoods are reported in the Appendix. One potential criticism is that 10 particles are not sufficient to well describe the parameter manifold. We thus conduct extra experiments, where instead of using particles,

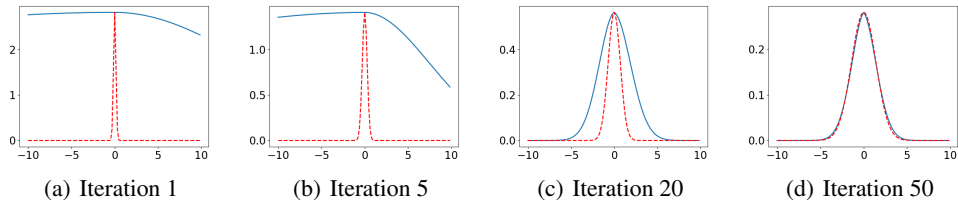

| (a) Iteration 1 | (b) Iteration 5 | (c) Iteration 20 | (d) Iteration 50 |

Figure 1: Evolution of the $a_{\mathcal{M}}^t p_{\phi}^t$ (blue solid line) and true heat kernel $k_{\mathcal{M}}^t$ (red dashed line) on $R$.

Table 1: Average test RMSE ($\downarrow$) for UCI regression.

|  | COMBINED | CONCRETE | KIN8NM | PROTEIN | WINE | YEAR |
|---|---|---|---|---|---|---|
| SVGD | $4.088 \pm 0.033$ | $5.027 \pm 0.116$ | $0.093 \pm 0.001$ | $4.186 \pm 0.017$ | $0.645 \pm 0.009$ | $8.686 \pm 0.010$ |
| HK-SVGD (OURS) | $\mathbf{4.077 \pm 0.035}$ | $\mathbf{4.814 \pm 0.112}$ | $\mathbf{0.091 \pm 0.001}$ | $\mathbf{4.138 \pm 0.019}$ | $\mathbf{0.624 \pm 0.010}$ | $\mathbf{8.656 \pm 0.007}$ |
| MSVGD-A | $4.056 \pm 0.033$ | $4.869 \pm 0.124$ | $0.092 \pm 0.001$ | $3.997 \pm 0.018$ | $0.637 \pm 0.008$ | $8.637 \pm 0.005$ |
| MSVGD-M | $4.029 \pm 0.033$ | $4.721 \pm 0.111$ | $0.090 \pm 0.001$ | $3.852 \pm 0.014$ | $0.637 \pm 0.009$ | $8.594 \pm 0.009$ |
| HK-MSVGD-A (OURS) | $4.020 \pm 0.043$ | $\mathbf{4.443 \pm 0.138}$ | $0.090 \pm 0.001$ | $4.001 \pm 0.004$ | $\mathbf{0.614 \pm 0.007}$ | $8.590 \pm 0.010$ |
| HK-MSVGD-M (OURS) | $\mathbf{3.998 \pm 0.046}$ | $4.552 \pm 0.146$ | $\mathbf{0.089 \pm 0.001}$ | $\mathbf{3.762 \pm 0.015}$ | $0.629 \pm 0.008$ | $\mathbf{8.533 \pm 0.005}$ |

we use a 2-layer neural network to generate parameter samples for BNN. The results are also reported in the Appendix. Consistently, better performance is achieved.

## 4.3 Deep Generative Models

Finally, we apply our framework to high-quality image generation. Four datasets are used in this experiment: CIFAR-10, STL-10, ImageNet, CelebA. Following [26, 29], images are scaled to the resolution of $32 \times 32, 48 \times 48, 64 \times 64$ and $160 \times 160$ respectively. Following [26, 27], we test 2 architectures on CIFAR-10 and STL-10, 1 architecture on CelebA and ImageNet. We report the standard Fréchet Inception Distance (FID) [31] and Inception Score (IS) [32] for evaluation. Architecture details and some experiments settings can be found in Appendix.

We compared our method with some popular and state-of-the-art GAN models under the same experimental setting, including WGAN-GP [33], SN-GAN[27], SMMD-GAN, SN-SMMD-GAN [26], CR-GAN [34], MMD-GAN with repulsive loss [29], Auto-GAN [35]. The results are reported in Table 2 where HK and HK-DK represent our model with kernel (1) and (2). More results are provided in the

Table 2: Results on image generation.

|  | CELEBA | | IMAGENET | |
|---|---|---|---|---|
|  | FID ($\downarrow$) | IS ($\uparrow$) | FID ($\downarrow$) | IS ($\uparrow$) |
| WGAN-GP | $29.2 \pm 0.2$ | $2.7 \pm 0.1$ | $65.7 \pm 0.3$ | $7.5 \pm 0.1$ |
| SN-GAN | $22.6 \pm 0.1$ | $2.7 \pm 0.1$ | $47.5 \pm 0.1$ | $11.2 \pm 0.1$ |
| SMMD-GAN | $18.4 \pm 0.2$ | $2.7 \pm 0.1$ | $38.4 \pm 0.3$ | $10.7 \pm 0.2$ |
| SN-SMMD-GAN | $12.4 \pm 0.2$ | $2.8 \pm 0.1$ | $36.6 \pm 0.2$ | $10.9 \pm 0.1$ |
| REPULSIVE | $10.5 \pm 0.1$ | $2.8 \pm 0.1$ | $31.0 \pm 0.1$ | $11.5 \pm 0.1$ |
| HK (OURS) | $\mathbf{9.7 \pm 0.1}$ | $\mathbf{2.9 \pm 0.1}$ | $29.6 \pm 0.1$ | $11.8 \pm 0.1$ |
| HK-DK (OURS) | – | – | $\mathbf{29.2 \pm 0.1}$ | $\mathbf{11.9 \pm 0.1}$ |

|  | CIFAR-10 | | STL-10 | |
|---|---|---|---|---|
|  | FID ($\downarrow$) | IS ($\uparrow$) | FID ($\downarrow$) | IS ($\uparrow$) |
|  | DC-GAN ARCHITECTURE | | | |
| WGAN-GP | $31.1 \pm 0.2$ | $6.9 \pm 0.2$ | $55.1$ | $8.4 \pm 0.1$ |
| SN-GAN | $25.5$ | $7.6 \pm 0.1$ | $43.2$ | $8.8 \pm 0.1$ |
| SMMD-GAN | $31.5 \pm 0.4$ | $7.0 \pm 0.1$ | $43.7 \pm 0.2$ | $8.4 \pm 0.1$ |
| SN-SMMD-GAN | $25.0 \pm 0.3$ | $7.3 \pm 0.1$ | $40.6 \pm 0.1$ | $8.5 \pm 0.1$ |
| CR-GAN | $18.7$ | $7.9$ | – | – |
| REPULSIVE | $16.7$ | $8.0$ | $36.7$ | $9.4$ |
| HK (OURS) | $14.9 \pm 0.1$ | $8.2 \pm 0.1$ | $31.8 \pm 0.1$ | $\mathbf{9.6 \pm 0.1}$ |
| HK-DK (OURS) | $\mathbf{13.2 \pm 0.1}$ | $\mathbf{8.4 \pm 0.1}$ | $\mathbf{30.3 \pm 0.1}$ | $\mathbf{9.6 \pm 0.1}$ |
|  | RESNET ARCHITECTURE | | | |
| SN-GAN | $21.7 \pm 0.2$ | $8.2 \pm 0.1$ | $40.1 \pm 0.5$ | $9.1 \pm 0.1$ |
| CR-GAN | $14.6$ | $8.4$ | – | – |
| REPULSIVE | $12.2 \pm 0.1$ | $8.3 \pm 0.1$ | $25.3 \pm 0.1$ | $10.2 \pm 0.1$ |
| AUTO-GAN | $12.4$ | $8.6 \pm 0.1$ | $31.1$ | $9.2 \pm 0.1$ |
| HK (OURS) | $11.5 \pm 0.1$ | $8.4 \pm 0.1$ | $24.3 \pm 0.1$ | $10.5 \pm 0.1$ |
| HK-DK (OURS) | $\mathbf{10.3 \pm 0.1}$ | $\mathbf{8.6 \pm 0.1}$ | $\mathbf{24.0 \pm 0.1}$ | $\mathbf{10.5 \pm 0.1}$ |
|  | BIGGAN SETTING | | | |
| BIGGAN | $14.7$ | – | – | – |
| CR-BIGGAN | $11.7$ | – | – | – |

Appendix. HK-DK exhibits some convergence issues on CelebA, hence no result is reported. We can see that our models achieve the state-of-the-art results (under the same experimental setting such as the same/similar architectures). Furthermore, compared to Auto-GAN, which needs 43 hours to train on the CIFAR-10 dataset due to the expensive architecture search, our method (HK with ResNet architecture) needs only 12 hours to obtain better results. Some randomly generated images are also provided in the Appendix.

## 5 Conclusion

We introduce the concept of implicit manifold learning, which implicitly learns the geometric information of an unknown manifold by learning the corresponding heat kernel. Both theoretical analysis and practical algorithm are derived. Our framework is flexible and can be applied to general kernel-based models, including DGMs and Bayesian inference. Extensive experiments suggest that our methods achieve consistently better results on different tasks, compared to related methods.

## Broader Impact

We propose a fundamentally novel method to implicitly learn the geometric information of a manifold by explicitly learning its associated heat kernel, which is the solution of heat equation with initial conditions given. Our proposed method is general and can be applied in many down-stream applications. Specifically, it could be used to improve many kernel-related algorithms and applications. It may also inspire researchers in deep learning to borrow ideas from other fields (mathematics, physics, etc.) and apply them to their own research. This can benefit both fields and thus promote interdisciplinary research.

## Acknowledgements

The research of the first and third authors was supported in part by NSF through grants CCF-1716400 and IIS-1910492.

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
