[Supplementary Material]

# A  Riemannian Manifold

**Definition 3 (Manifold)** *[36] Let $\mathcal{M}$ be a set. If there exists a set of coordinate systems $A$ for $\mathcal{M}$ satisfying the conditions below, $(\mathcal{M}, A)$ is called an $n$-dimensional $C^\infty$ differentiable manifold, or simply manifold.*

1. *Each element $\phi$ of $A$ is a one-to-one mapping from $\mathcal{M}$ to some open subset of $R^n$;*

2. *For all $\phi \in A$, given any one-to-one mapping $\psi$ from $\mathcal{M}$ to $R^n$, the following holds:*
$$\psi \in A \Leftrightarrow \psi \circ \phi^{-1} \text{is a } C^\infty \text{ diffeomorphism};$$

By $C^\infty$ diffeomorphism, we mean that $\psi \circ \phi^{-1}$ and its inverse $\phi \circ \psi^{-1}$ are both $C^\infty$ (infinitely many times differentiable). Infinitely differentiable is not necessary actually, we may consider this notation as 'sufficiently smooth'.

We will use $T_\mathbf{x}\mathcal{M}$ to denote the tangent space of $\mathcal{M}$ at point $\mathbf{x}$, and $X, Y, Z$ to denote the vector fields.

**Definition 4 (Riemannian Metric and Riemannian Manifold)** *[37] Let $\mathcal{M}$ be a manifold, $C^\infty(\mathcal{M})$ be the comminicative ring of smooth functions on $\mathcal{M}$, and $C^\infty(\mathcal{TM})$ be the set of smooth vector fields on $\mathcal{M}$ forming a module over $C^\infty(\mathcal{M})$. A Riemannian metric $g$ on $\mathcal{M}$ is a tensor field $g : C^\infty(\mathcal{TM}) \otimes C^\infty(\mathcal{TM}) \to C^\infty(\mathcal{M})$ such that for each $\mathbf{x} \in \mathcal{M}$, the restriction $g_\mathbf{x}$ of $g$ to the tensor product $T_\mathbf{x}\mathcal{M} \otimes T_\mathbf{x}\mathcal{M}$ with:*
$$g_\mathbf{x} : (X_\mathbf{x}, Y_\mathbf{x}) \to g(X, Y)(\mathbf{x})$$

*is a real scalar product on the tangent space $T_\mathbf{x}\mathcal{M}$. The pair $(\mathcal{M}, g)$ is called a Riemannian manifold. The geometric properties of $(\mathcal{M}, g)$ which depend only on the metric $g$ are said to be intrinsic or metric properties.*

One classical example is that the Riemannian manifold $E^m = (R^m, \langle,\rangle_{R^m})$ is nothing but the m-dimensional Euclidean space.

The Riemannian curvature tensor of a manifold $\mathcal{M}$ is defined by
$$R(X, Y)Z = \nabla_X \nabla_Y Z - \nabla_Y \nabla_X Z - \nabla_{[X,Y]} Z$$

on vector fields $X, Y, Z$. For any two tangent vector $\xi, \eta \in T_\mathbf{x}\mathcal{M}$, we use $Ric_\mathbf{x}(\xi, \eta)$ to denote the Ricci tensor evaluated at $(\xi, \eta)$, which is defined to be the trace of the mapping $T_\mathbf{x}\mathcal{M} \to T_\mathbf{x}\mathcal{M}$ given by $\zeta \to R(\zeta, \eta)\xi$.

We use $Ric \geq K$ to denote that a manifold's Ricci curvature is bounded from below by $K$, in the sense that $Ric_\mathbf{x}(\xi, \xi) \geq K|\xi|^2$ for all $\mathbf{x} \in \mathcal{M}, \xi \in T_\mathbf{x}\mathcal{M}$.

# B  Proof of Theorem 3

**Theorem 3** *Let $(\mathcal{M}, g)$ be a connected, complete Riemannian manifold with Riemannian volume element $\mathrm{d}m$ and positive Ricci curvature. Assume that the heat kernel is Lipschitz continous. Let $p^t, q^t$ be two evolving PDFs induced by WGF for negative entropy as defined in Definition 2, their corresponding probability measure are $\boldsymbol{\mu}^t$ and $\boldsymbol{\nu}^t$. If $d_W^2(\boldsymbol{\mu}^0, \boldsymbol{\nu}^0) < \infty$, then $p^t(\mathbf{x}) = q^t(\mathbf{x})$ almost everywhere as $t \to \infty$; furthermore, $\int_{\mathbf{x} \in \mathcal{M}} \|p^t(\mathbf{x}) - q^t(\mathbf{x})\|^2 \mathrm{d}m$ converges to 0 exponentially fast.*

**Proof**  We start by introducing the following lemma, which is Proposition 4.4 in [20].

**Lemma 8 ([20])** *Assume $Ric \geq K$. Let $(\boldsymbol{\mu}_\mathcal{M}^t)_{t \geq 0}$ and $(\boldsymbol{\nu}_\mathcal{M}^t)_{t \geq 0}$ be two trajectories of the gradient flow of negative entropy functional with initial distribution $\boldsymbol{\mu}_\mathcal{M}^0$ and $\boldsymbol{\nu}_\mathcal{M}^0$, respectively, then*
$$d_W^2(\boldsymbol{\mu}_\mathcal{M}^t, \boldsymbol{\nu}_\mathcal{M}^t) \leq e^{-Kt} d_W^2(\boldsymbol{\mu}_\mathcal{M}^0, \boldsymbol{\nu}_\mathcal{M}^0).$$

*In particular, for a given initial value $\boldsymbol{\mu}_\mathcal{M}^0$, there is at most one trajectory of the gradient flow.*

What we need to do is to extend the results of Lemma 8 from probability measures to probability density functions. Following some previous work, we first define the projection operator.

**Definition 5 (Projection Operator)** *[8] Let $\pi^i, \pi^{i,j}$ denote the projection operators defined on the product space $\bar{\mathbf{X}} := X_1 \times ... \times X_N$ respectively such that:*

$$\pi^i : (\mathbf{x}_1, ..., \mathbf{x}_N) \mapsto \mathbf{x}_i \in X_i, \ \pi^{i,j} : (\mathbf{x}_1, ..., \mathbf{x}_N) \mapsto (\mathbf{x}_i, \mathbf{x}_j) \in X_i \times X_j$$

*if $\bar{\mu} \in \mathcal{P}(\bar{\mathbf{X}})$, the marginals of $\bar{\mu}$ are the probability measures*

$$\boldsymbol{\mu}^i := \pi^i_{\#}\bar{\boldsymbol{\mu}} \in \mathcal{P}(X_i), \ \boldsymbol{\mu}^{i,j} := \pi^{i,j}_{\#}\bar{\boldsymbol{\mu}} \in \mathcal{P}(X_i \times X_j).$$

We then introduce the following lemma which utilize the projection operator.

**Lemma 9** *[8] Let $X_i$, $i \in \mathbb{N}$, be a sequence of Radon separable metric spaces, $\boldsymbol{\mu}_i \in \mathcal{P}(X_i)$ and $\alpha^{i(i+1)} \in \Gamma(\boldsymbol{\mu}^i, \boldsymbol{\mu}^{i+1})$, $\beta^{1i} \in \Gamma(\boldsymbol{\mu}^1, \boldsymbol{\mu}^i)$, where $\Gamma(\boldsymbol{\mu}, \boldsymbol{\nu})$ denotes the 2-plan (i.e. transportation plan between two distributions) with given marginals $\boldsymbol{\mu}, \boldsymbol{\nu}$. Let $\bar{\mathbf{X}}_\infty := \prod_{i \in \mathbb{N}} X_i$, with the canonical product topology. Then there exist $\bar{\nu}, \bar{\mu} \in \mathcal{P}(\bar{\mathbf{X}}_\infty)$ such that*

$$\pi^{i,i+1}_{\#}\bar{\boldsymbol{\mu}} = \alpha^{i(i+1)}, \ \pi^{1,i}_{\#}\bar{\boldsymbol{\nu}} = \beta^{1i}, \ \forall i \in \mathbb{N}. \tag{9}$$

Now we are ready to prove the theorem.

We first construct a sequence of probability measures $\{\boldsymbol{\rho}^t\}_{t \in \mathbb{N}}$, such that $\boldsymbol{\rho}_{2t} = \boldsymbol{\mu}^t_{\boldsymbol{\phi}}$, $\boldsymbol{\rho}^{2t+1} = \boldsymbol{\mu}^t_{\mathcal{M}}$, $t \in \mathbb{N}$. Then, we choose $\alpha^{t(t+1)} \in \Gamma_o(\boldsymbol{\rho}^t, \boldsymbol{\rho}^{t+1})$, i.e. the optimal 2-plans given marginal $\boldsymbol{\rho}^t, \boldsymbol{\rho}^{t+1}$. According to Lemma 9, we can find a probability measure $\bar{\boldsymbol{\mu}} \in \mathcal{P}(\bar{\mathcal{M}}_\infty)$ where $\bar{\mathcal{M}}_\infty := \prod_{t \in \mathbb{N}} \mathcal{M}_t$ satisfying (9). Then the 2-Wasserstein distance can be written as:

$$d^2_W(\boldsymbol{\mu}^t_{\boldsymbol{\phi}}, \boldsymbol{\mu}^t_{\mathcal{M}}) = d^2_W(\boldsymbol{\rho}^{2t}, \boldsymbol{\rho}^{2t+1}) = \boldsymbol{d}(\pi^{2t}, \pi^{2t+1})_{L^2(\bar{\boldsymbol{\mu}}; \bar{\mathcal{M}}_\infty)},$$

where $\boldsymbol{d}(\cdot, \cdot)_{L^2(\bar{\boldsymbol{\mu}}; \bar{\mathcal{M}}_\infty)}$ denotes the norm of $L^2$ space.

According to Lemma 8,

$$d^2_W(\boldsymbol{\mu}^t_{\boldsymbol{\phi}}, \boldsymbol{\mu}^t_{\mathcal{M}}) \le e^{-Kt} d^2_W(\boldsymbol{\mu}^0_{\boldsymbol{\phi}}, \boldsymbol{\mu}^0_{\mathcal{M}}),$$

which means:

$$\boldsymbol{d}(\pi^{2t}, \pi^{2t+1})_{L^2(\bar{\boldsymbol{\mu}}; X)} \le e^{-Kt} d^2_W(\boldsymbol{\mu}^0_{\boldsymbol{\phi}}, \boldsymbol{\mu}^0_{\mathcal{M}}).$$

$$
\begin{aligned}
\int_{\mathcal{M}} \|k^t_{\boldsymbol{\phi}} - k^t_{\mathcal{M}}\|^2 \mathrm{d}m &= \int_{\mathcal{M}} (k^t_{\boldsymbol{\phi}})^2 \mathrm{d}m + \int_{\mathcal{M}} (k^t_{\mathcal{M}})^2 \mathrm{d}m - 2\int_{\mathcal{M}} k^t_{\mathcal{M}} k^t_{\boldsymbol{\phi}} \mathrm{d}m \\
&\le \int_{\mathcal{M}} k^t_{\boldsymbol{\phi}} \mathrm{d}\mu^t_{\boldsymbol{\phi}} + \int_{\mathcal{M}} k^t_{\mathcal{M}} \mathrm{d}\mu^t_{\mathcal{M}} \\
&= \int_{\bar{\mathcal{M}}_\infty} (k^t_{\boldsymbol{\phi}} \circ \pi^{2t}) \mathrm{d}\bar{\boldsymbol{\mu}} + \int_{\bar{\mathcal{M}}_\infty} (k^t_{\mathcal{M}} \circ \pi^{2t+1}) \mathrm{d}\bar{\boldsymbol{\mu}} \\
&\le \int_{\bar{\mathcal{M}}_\infty} \|k^t_{\boldsymbol{\phi}} \circ \pi^{2t} + k^t_{\mathcal{M}} \circ \pi^{2t+1}\| \mathrm{d}\bar{\boldsymbol{\mu}} \\
&\le (\int_{\bar{\mathcal{M}}_\infty} \|k^t_{\boldsymbol{\phi}} \circ \pi^{2t} + k^t_{\mathcal{M}} \circ \pi^{2t+1}\|^2 \mathrm{d}\bar{\boldsymbol{\mu}})^{1/2}, \text{ by Jensen's Inequality} \\
&\le (\int_{\bar{\mathcal{M}}_\infty} L^2_{sup} \|\pi^{2t} - \pi^{2t+1}\|^2 \mathrm{d}\bar{\boldsymbol{\mu}})^{1/2} \\
&= C\boldsymbol{d}(\pi^{2t}, \pi^{2t+1})_{L^2(\bar{\boldsymbol{\mu}}; \bar{\mathcal{M}}_\infty)}, \text{C is a constant because of Lipschitz continuity} \\
&\le C e^{-Kt} d^2_W(\boldsymbol{\mu}^0_{\boldsymbol{\phi}}, \boldsymbol{\mu}^0_{\mathcal{M}}).
\end{aligned}
$$

which completes the proof. Readers who are interested may also refer to Chapter 5.3 in [8] and proof of Proposition 4.4 in [20] for more information. ∎

## C  Proof of Theorem 4 and Theorem 5

**Theorem 5** *Let $(\mathcal{M}, g)$ be a complete Riemannian manifold without boundary or compact Riemannian manifold with convex boundary $\partial\mathcal{M}$. If $\mathcal{M}$ has positive Ricci curvature bounded by $K$ and its dimension $dim(\mathcal{M}) \geq 1$. Let $k_{\mathcal{M}}^t(\mathbf{x})$ be the heat kernel of*

$$\frac{\partial k_{\mathcal{M}}(t, \mathbf{x}_0, \mathbf{x})}{\partial t} = \triangle_{\mathbf{x}} k_{\mathcal{M}}(t, \mathbf{x}_0, \mathbf{x}), \frac{\partial k_{\mathcal{M}}^t}{\partial n} = 0 \text{ on } \partial\mathcal{M} \text{ if applicable}$$

*where $n$ denotes the outward-pointing unit normal to boundary $\partial\mathcal{M}$. Then*

$$k_{\mathcal{M}}^t(\mathbf{x}) \geq \frac{\Gamma(dim(\mathcal{M})/2 + 1)}{C(\epsilon)(\pi t)^{dim(\mathcal{M})/2}} \exp(\frac{\pi^2 - \pi^2 dim(\mathcal{M})}{(4 - \epsilon)Kt})$$

*for $\forall \mathbf{x}_0, \mathbf{x} \in \mathcal{M}$ and small $\epsilon > 0$, where $C(\epsilon)$ is a constant depending on $\epsilon > 0$ and $d$ such that $C(\epsilon) \to 0$ as $\epsilon \to 0$, and $\Gamma$ is the gamma function.*

**Proof**  We start by introducing the following lemma:

**Lemma 10** *[21] Let $\mathcal{M}$ be a complete Riemannian manifold without boundary or compact Riemannian manifold with convex boundary $\partial\mathcal{M}$. Suppose that the Ricci curvature of $\mathcal{M}$ is non-negative. Let $k_{\mathcal{M}}^t(\mathbf{x}) = k_{\mathcal{M}}(t, \mathbf{x}_0, \mathbf{x})$ be the heat kernel of*

$$\frac{\partial k_{\mathcal{M}}(t, \mathbf{x}_0, \mathbf{x})}{\partial t} = \triangle_{\mathbf{x}} k_{\mathcal{M}}(t, \mathbf{x}_0, \mathbf{x}), \frac{\partial k_{\mathcal{M}}^t}{\partial n} = 0 \text{ on } \partial\mathcal{M} \text{ if applicable}$$

*Then, the heat kernel satisfies*

$$k_{\mathcal{M}}(t, \mathbf{x}_0, \mathbf{x}) \geq C^{-1}(\epsilon) V^{-1}[B_{\mathcal{M}, \mathbf{x}}(\sqrt{t})] \exp(\frac{-d_{\mathcal{M}}^2(\mathbf{x}_0, \mathbf{x})}{(4 - \epsilon)t})$$

*for some constant $C(\epsilon)$ depending on $\epsilon > 0$ and $n$ such that $C(\epsilon) \to 0$ as $\epsilon \to 0$, where $B_{\mathcal{M}, \mathbf{x}}(\sqrt{t}) \subseteq \mathcal{M}$, denotes the geodesic ball with radius $\sqrt{t}$ around $\mathbf{x}$. Moreover, by symmetrizing,*

$$k_{\mathcal{M}}(t, \mathbf{x}_0, \mathbf{x}) \geq C^{-1}(\epsilon) V^{-1/2}[B_{\mathcal{M}, \mathbf{x}_0}(\sqrt{t})] V^{-1/2}[B_{\mathcal{M}, \mathbf{x}}(\sqrt{t})] \exp(\frac{-d_{\mathcal{M}}^2(\mathbf{x}_0, \mathbf{x})}{(4 - \epsilon)t}).$$

Lemma 10 comes from the Theorem 4.1 and Theorem 4.2 in [21]. Then we need to introduce the definition of CD condition and Lemma 11.

**Definition 6 (CD condition)** *[38] For Rimennian manifold $\mathcal{M}$, the curvature-dimension (CD) condition $CD(K, d)$ is satisfied if and only if the dimension of manifold is less or equal to $d$ and Ricci curvature is bounded from below by $K$.*

**Lemma 11** *[38] For every metric measure space $(\mathcal{M}, d_{\mathcal{M}}, m)$ which satisfies the curvature-dimension condition $CD(K, d)$ for some real numbers $K > 0$ and $d \geq 1$, the support of $m$ is compact and has diameter*

$$L \leq \pi\sqrt{\frac{d - 1}{K}},$$

*where the diameter is defined as*

$$L = \sup_{\mathbf{x}, \mathbf{y} \in \mathcal{M} \times \mathcal{M}} d_{\mathcal{M}}(\mathbf{x}, \mathbf{y}).$$

Now we are ready to prove the Theorem.

Let $(\mathcal{M}, g)$ be a complete Riemannian manifold with positive Ricci curvature $Ric \geq K$ and dimension $dim(\mathcal{M})$, then

$$Ric \geq 0 = [dim(\mathcal{M}) - 1] \cdot 0$$

Because Euclidean Space can be seen as a manifold with constant sectional curvature 0. By Bishop-Gromov inequality we have the following for manifold with non-negative Ricci curvature:

$$V[B_{\mathcal{M}, \mathbf{x}}(r)] \leq V[B_{R^{dim(\mathcal{M})}}(r)]$$

i.e. the volume of a geodesic ball with radius $r$ around $\mathbf{x}$ is less than the ball with same radius in a $dim(\mathcal{M})$-dimensional Euclidean space. According to the volume of n-ball, we have

$$V[B_{R^{dim(\mathcal{M})}}(r)] = \frac{\pi^{dim(\mathcal{M})/2}}{\Gamma(dim(\mathcal{M})/2 + 1)} r^{dim(\mathcal{M})}$$

Thus we have

$$V^{-1}[B_{\mathcal{M},\mathbf{x}}(r)] \geq \frac{\Gamma(dim(\mathcal{M})/2 + 1)}{\pi^{dim(\mathcal{M})/2}} r^{-dim(\mathcal{M})}$$

Using Lemma 10, we have:

$$k_{\mathcal{M}}(t,\mathbf{x}_0,\mathbf{x}) \geq C^{-1}(\epsilon) \frac{\Gamma(dim(\mathcal{M})/2 + 1)}{\pi^{dim(\mathcal{M})/2} t^{dim(\mathcal{M})/2}} \exp(\frac{-d_{\mathcal{M}}^2(\mathbf{x}_0,\mathbf{x})}{(4-\epsilon)t})$$

Using Lemma 11 and $d_{\mathcal{M}}(\mathbf{x}_0,\mathbf{x}) \leq L \leq \pi\sqrt{\frac{dim(\mathcal{M}) - 1}{K}}$, we have:

$$k_{\mathcal{M}}(t,\mathbf{x}_0,\mathbf{x}) \geq \frac{\Gamma(dim(\mathcal{M})/2 + 1)}{C(\epsilon)\pi^{dim(\mathcal{M})/2} t^{dim(\mathcal{M})/2}} \exp(\frac{\pi^2 - \pi^2 dim(\mathcal{M})}{(4-\epsilon)Kt})$$

We now can conclude Theorem 5. The term $\exp(\dfrac{\pi^2 - \pi^2 dim(\mathcal{M})}{(4-\epsilon)Kt})$ is increasing, and $\dfrac{\Gamma(dim(\mathcal{M})/2 + 1)}{C(\epsilon)\pi^{dim(\mathcal{M})/2} t^{dim(\mathcal{M})/2}}$ is decreasing polynomially. We can see that the lower bound of heat kernel decrease to 0 polynomially with respect to t. Thus heat kernel value decrease to 0 at most polynomially with respect to t. ∎

**Theorem 4** *Let $(\mathcal{M}, g)$ be a complete Riemannian manifold without boundary or compact Riemannian manifold with convex boundary $\partial\mathcal{M}$. Assume it has positive Ricci curvature. $\mathrm{d}m$ denotes its Riemannian volume element. Let $k_{\mathcal{M}}^t(\mathbf{x})$ be the heat kernel of*

$$\frac{\partial k_{\mathcal{M}}(t,\mathbf{x}_0,\mathbf{x})}{\partial t} = \triangle_{\mathbf{x}} k_{\mathcal{M}}(t,\mathbf{x}_0,\mathbf{x}), \frac{\partial k_{\mathcal{M}}^t}{\partial n} = 0 \text{ on } \partial\mathcal{M} \text{ if applicable}$$

*where $n$ denotes the outward-pointing unit normal to boundary $\partial\mathcal{M}$. Then $\int_{\mathbf{x}\in\mathcal{M}} \|k_{\mathcal{M}}^t(\mathbf{x})\|^2 \mathrm{d}m$ converges to 0 at most polynomially as $t \to \infty$, which is slower than $\int_{\mathbf{x}\in\mathcal{M}} \|p^t(\mathbf{x}) - k_{\mathcal{M}}^t(\mathbf{x})\|^2 \mathrm{d}m$.*

**Proof** For a given manifold $\mathcal{M}$, its Riemannian volume element doesn't vary at different time t, thus the lower bound of integral $\int_{\mathbf{x}\in\mathcal{M}} \|k_{\mathcal{M}}^t(\mathbf{x})\|^2 \mathrm{d}m$ where $k_{\mathcal{M}}^t(\mathbf{x}) = k_{\mathcal{M}}(t,\mathbf{x}_0,\mathbf{x})$ also decrease polynomially because of Theorem 5. Then we can conclude Theorem 4. ∎

# D  MMD and SMMD

In our proposed method for deep generative models, MMD is used as the objective function for the generator, which is defined as:

$$\begin{aligned}\mathsf{MMD}^2(\mathbb{P},\mathbb{Q}) =& \mathbb{E}_{\mathbf{x}_i,\mathbf{x}_j\sim\mathbb{P}}(k_{\boldsymbol\phi}(\mathbf{x}_i,\mathbf{x}_j)) + \mathbb{E}_{\mathbf{y}_i,\mathbf{y}_j\sim\mathbb{Q}}(k_{\boldsymbol\phi}(\mathbf{y}_i,\mathbf{y}_j)) \\ &- 2\mathbb{E}_{\mathbf{x}_i\sim\mathbb{P},\mathbf{y}_j\sim\mathbb{Q}}(k_{\boldsymbol\phi}(\mathbf{x}_i,\mathbf{y}_j)),\end{aligned} \tag{10}$$

where $\mathbb{P}, \mathbb{Q}$ are probability distributions of the training data and the generated data, respectively. MMD measures the difference between two distributions. Thus, we want it to be minimized.

Generator can also use SMMD [26] as the objective function, which is defined as:

$$\mathsf{SMMD}^2(\mathbb{P},\mathbb{Q}) = \sigma \mathsf{MMD}^2(\mathbb{P},\mathbb{Q})$$

$$\sigma = \{\zeta + \mathbb{E}_{\mathbf{x}\in\mathbb{P}}[k_{\boldsymbol\phi}(t,\mathbf{x},\mathbf{x})] + \sum_{i=1}^{d}\mathbb{E}_{\mathbf{x}\in\mathbb{P}}\left[\frac{\partial^2 k_{\boldsymbol\phi}(t,\mathbf{y},\mathbf{z})}{\partial\mathbf{y}_i \partial\mathbf{z}_i}|_{(\mathbf{y},\mathbf{z})=(\mathbf{x},\mathbf{x})}\right]\}^{-1}, \tag{11}$$

where $d$ is the dimensionality of the data, $\mathbf{y}_i$ denotes the $i^{th}$ element of $\mathbf{y}$, and $\zeta$ is a hyper-parameter.

# E Algorithms

## E.1 Algorithms for DGM with heat kernel learning

First of all, let's introduce a similar objective function for kernel with form (2), which can be used to replace (7).

Similarly to (6), for kernels with form (2), we have the following bound:

$$\|p_{\boldsymbol{\phi}}^t(\mathbf{y},\mathbf{x}) - p_{\boldsymbol{\phi}}^t(\mathbf{y},\mathbf{z})\| \leq c_1 \|\mathbb{E}\{\sin\{(\boldsymbol{\omega}_{\boldsymbol{\psi}_1}^t)^\intercal [h_{\boldsymbol{\phi}}^t(\mathbf{y}) - h_{\boldsymbol{\phi}}^t(\mathbf{x})]\}\}\| \|\boldsymbol{\omega}_{\boldsymbol{\psi}_1}^t\| \|\nabla_{\mathbf{x}} h_{\boldsymbol{\phi}}^t(\mathbf{x})\|_{\mathcal{F}}$$
$$+ c_2 \|\mathbb{E}\{\sin\{(\boldsymbol{\omega}_{\boldsymbol{\psi}_2,\mathbf{x},\mathbf{y}}^t)^\intercal [h_{\boldsymbol{\phi}}^t(\mathbf{y}) - h_{\boldsymbol{\phi}}^t(\mathbf{x})]\}\}\| \|\boldsymbol{\omega}_{\boldsymbol{\psi}_2,\mathbf{x},\mathbf{y}}^t\| \|\nabla_{\mathbf{x}} h_{\boldsymbol{\phi}}^t(\mathbf{x})\|_{\mathcal{F}} \tag{12}$$

where $c_1 = \|\mathbf{x} - \mathbf{z}\|$, $c_2 = \|\mathbf{x} - \mathbf{z}\| \left[ \|\boldsymbol{\omega}_{\boldsymbol{\psi}_2,\mathbf{x},\mathbf{y}}^t\| + \|h_{\boldsymbol{\phi}}^t(\mathbf{y}) - h_{\boldsymbol{\phi}}^t(\mathbf{x})\| \|\frac{\partial \boldsymbol{\omega}_{\boldsymbol{\psi}_2,\mathbf{x},\mathbf{y}}^t}{\partial h_{\boldsymbol{\phi}}^t(\mathbf{x})}\|_{\mathcal{F}} \right]$

Incorporating this bound into objective as (7), the optimization problem for learning kernel with form (2) becomes:

$$\min_{\boldsymbol{\phi}} \alpha H(\tilde{\boldsymbol{\mu}}_{\boldsymbol{\phi}}^t) + \beta d_W^2(\boldsymbol{\nu}, \tilde{\boldsymbol{\mu}}_{\boldsymbol{\phi}}^t) - \lambda \mathbb{E}_{\mathbf{y} \neq \mathbf{x}} \left[ p_{\boldsymbol{\phi}}^t(\mathbf{y},\mathbf{x}) \right] \tag{13}$$
$$+ \mathbb{E}_{\mathbf{x} \sim \mathbb{P}, \mathbf{y} \sim \mathbb{Q}} \left[ \gamma_1 k_{sin}(t,\mathbf{y},\mathbf{x}) + \gamma_2 \|h_{\boldsymbol{\phi}}^t(\mathbf{x}) - h_{\boldsymbol{\phi}}^t(\mathbf{y})\| \right]$$
$$+ \mathbb{E}_{\mathbf{x}_i,\mathbf{x}_j \sim \mathbb{P}} \left[ \gamma_3 k_{sin}(t,\mathbf{x}_j,\mathbf{x}_i) + \gamma_4 \|h_{\boldsymbol{\phi}}^t(\mathbf{x}_i) - h_{\boldsymbol{\phi}}^t(\mathbf{x}_j)\| \right]$$
$$+ \gamma_5 \mathbb{E}_{\mathbf{x} \sim \mathbb{P}, \mathbf{y} \neq \mathbf{x}} \left[ \|\boldsymbol{\omega}_{\boldsymbol{\psi}_1}^t\| + \|\boldsymbol{\omega}_{\boldsymbol{\psi}_2,\mathbf{x},\mathbf{y}}^t\| + \|\frac{\partial \boldsymbol{\omega}_{\boldsymbol{\psi}_2,\mathbf{x},\mathbf{y}}^t}{\partial h_{\boldsymbol{\phi}}^t(\mathbf{x})}\|_{\mathcal{F}} \right],$$
$$\text{where } k_{sin}(t,\mathbf{y},\mathbf{x}) = \mathbb{E}\{\sin\{(\boldsymbol{\omega}_{\boldsymbol{\psi}_1}^t)^\intercal [h_{\boldsymbol{\phi}}^t(\mathbf{y}) - h_{\boldsymbol{\phi}}^t(\mathbf{x})]\}\}$$
$$+ \mathbb{E}\{\sin\{(\boldsymbol{\omega}_{\boldsymbol{\psi}_2,\mathbf{x},\mathbf{y}}^t)^\intercal [h_{\boldsymbol{\phi}}^t(\mathbf{y}) - h_{\boldsymbol{\phi}}^t(\mathbf{x})]\}\},$$

Now we present the algorithm of DGM with heat kernel learning here.

---

**Algorithm 2** Deep Generative Model with Heat Kernel Learning

---

**Input:** training data $\{\mathbf{x}_i\}$ on manifold $\mathcal{M}_{\mathcal{P}}$, generator $g_{\boldsymbol{\theta}}$, denote generated data as $\{\mathbf{y}_i\}$, kernel parameterized by (1) or (2), all the hyper-parameters in (7), time step $\tau = \alpha/2\beta$.
**for** training epochs $s$ **do**
    **for** iteration j **do**
        Sample $\{\mathbf{x}_i\}_{i=1}^n$ and $\{\mathbf{y}_i\}_{i=1}^n$.
        Initialize function $p_{\boldsymbol{\phi}}^0$, compute corresponding $\boldsymbol{\nu} = \tilde{\boldsymbol{\mu}}_{\boldsymbol{\phi}}^0$ by (3).
        **for** $k = 1$ **to** $m$ **do**
            Compute $\tilde{\boldsymbol{\mu}}_{\boldsymbol{\phi}}^{k\tau}$ by (3), solve (7) or (13). Update $\boldsymbol{\nu} \leftarrow \tilde{\boldsymbol{\mu}}_{\boldsymbol{\phi}}^{k\tau}$, where $\tilde{\boldsymbol{\mu}}_{\boldsymbol{\phi}}^{k\tau}$ is computed by (3).
        **end for**
    **end for**
    Sample $\{\mathbf{x}_i\}_{i=1}^n$ and $\{\mathbf{y}_i\}_{i=1}^n$. Update $\boldsymbol{\theta}$ by minimizing MMD computed with (1) or (2).
**end for**

---

## E.2 Some discussions

Instead of initializing the $\boldsymbol{\nu} = \tilde{\boldsymbol{\mu}}_{\boldsymbol{\phi}}^0$ by Equation (3), we can also simply initialize it to be $1/n$, which represents the discrete uniform distribution. In this case, we set the $\boldsymbol{\mu}_{\boldsymbol{\phi}}^t$ for time $t$ to be

$$\boldsymbol{\mu}_{\boldsymbol{\phi}}^t(\mathbf{x}) = \frac{\sum_{j=1}^n p_{\boldsymbol{\phi}}^t(\mathbf{x}_j, \mathbf{x})}{n \sum_{j=1}^n p_{\boldsymbol{\phi}}^0(\mathbf{x}_j, \mathbf{x})},$$

where unknown constant $\alpha_{\mathcal{M}}^t$ is also cancelled.

To approximate $H(\tilde{\boldsymbol{\mu}}_{\boldsymbol{\phi}}^t)$, we may use either

$$H(\tilde{\boldsymbol{\mu}}_{\boldsymbol{\phi}}^t) \approx \sum_{i=1}^{n} \tilde{\boldsymbol{\mu}}_{\boldsymbol{\phi}}^t(\mathbf{x}_i) \log \tilde{\boldsymbol{\mu}}_{\boldsymbol{\phi}}^t(\mathbf{x}_i)$$

or

$$H(\tilde{\boldsymbol{\mu}}_{\boldsymbol{\phi}}^t) \approx \frac{1}{n} \sum_{j=1}^{n} \sum_{i=1}^{n} \tilde{\boldsymbol{\mu}}_{\boldsymbol{\phi}}^t(\mathbf{x}_i) \log p_{\boldsymbol{\phi}}^t(\mathbf{x}_j, \mathbf{x}_i).$$

In practice, these different implementations may need different hyper-parameter settings, and have different performances. Furthermore, we observed that using unnormalized density estimation $\boldsymbol{\mu}_{\boldsymbol{\phi}}^t$ instead of $\tilde{\boldsymbol{\mu}}_{\boldsymbol{\phi}}^t$ also leads to competitive results.

## F  Experimental Results and Settings on Improved SVGD

We provide some experimental settings here. Our implementation is based on TensorFlow with a Nvidia 2080 Ti GPU. To simplify the setting, the RBF-kernel is used in all layers except the last one, which is learned by our method (1) with $h_{\boldsymbol{\phi}}$ a 2-layer neural network. In other words, we only learn the parameter manifold for the last layer. Following [30], we run 20 trials on all the datasets except for Protein and Year, where 5 trials are used. At each trial, we randomly choose $90\%$ of the dataset as the training set, and the rest $10\%$ as the testing set. For large datasets like Year and Combined, we use the Adam optimizer with a batch size of 1000, and use batch size of 100 for all other datasets. Before every update of the BNN parameters, we run Algorithm 1 with $m = 1$; and 5 Adam update steps are implemented to solve (4). For matrix-valued SVGD, We use the same experimental setting, except that the number of update steps for solving (4) is chosen from $\{1, 2, 5, 10\}$, based on hyper-parameter tuning.

We report the average test log-likelihood in Table 3, from which we can also see that our proposed method improves model performance.

Table 3: Average test log-likelihood ($\uparrow$) for UCI regression.

|  | COMBINED | CONCRETE | KIN8NM | PROTEIN | WINE | YEAR |
|---|---|---|---|---|---|---|
| SVGD | $-2.832 \pm 0.009$ | $-3.064 \pm 0.034$ | $0.964 \pm 0.012$ | $-2.846 \pm 0.003$ | $-0.997 \pm 0.019$ | $-3.577 \pm 0.002$ |
| HK-SVGD (OURS) | $-2.827 \pm 0.009$ | $-3.015 \pm 0.037$ | $0.976 \pm 0.007$ | $-2.838 \pm 0.004$ | $-0.958 \pm 0.021$ | $-3.559 \pm 0.001$ |
| MSVGD-A | $-2.824 \pm 0.009$ | $-3.150 \pm 0.054$ | $0.956 \pm 0.011$ | $-2.796 \pm 0.004$ | $-0.980 \pm 0.016$ | $-3.569 \pm 0.001$ |
| MSVGD-M | $-2.817 \pm 0.009$ | $-3.207 \pm 0.071$ | $0.975 \pm 0.011$ | $-2.755 \pm 0.003$ | $-0.988 \pm 0.018$ | $-3.561 \pm 0.002$ |
| HK-MSVGD-A (OURS) | $-2.815 \pm 0.012$ | $\mathbf{-3.011 \pm 0.076}$ | $0.982 \pm 0.011$ | $-2.800 \pm 0.001$ | $\mathbf{-0.943 \pm 0.016}$ | $-3.549 \pm 0.002$ |
| HK-MSVGD-M (OURS) | $\mathbf{-2.814 \pm 0.013}$ | $-3.157 \pm 0.067$ | $\mathbf{0.989 \pm 0.009}$ | $\mathbf{-2.731 \pm 0.004}$ | $-1.013 \pm 0.019$ | $\mathbf{-3.534 \pm 0.001}$ |

Instead of using particles, we further improve our proposed HK-SVGD by introducing a parameter generator, which takes Gaussian noises as inputs and outputs samples of parameter distribution for BNNs. We use a 2-layer neural network to model this generator, 10 samples are generated at each iteration. We denote the resulting model as HK-ISVGD, and compare it with vanilla SVGD and our proposed HK-SVGD. Results on UCI regression are shown in Table 4 and Table 5. We can see that introducing the parameter sample generator will lead to performance improvement on most of the datasets.

Table 4: Average test RMSE ($\downarrow$) for UCI regression with parameter generator.

|  | COMBINED | CONCRETE | KIN8NM | PROTEIN | WINE | YEAR |
|---|---|---|---|---|---|---|
| SVGD | $4.088 \pm 0.033$ | $5.027 \pm 0.116$ | $0.093 \pm 0.001$ | $4.186 \pm 0.017$ | $0.645 \pm 0.009$ | $8.686 \pm 0.010$ |
| HK-SVGD (OURS) | $4.077 \pm 0.035$ | $\mathbf{4.814 \pm 0.112}$ | $0.091 \pm 0.001$ | $4.138 \pm 0.019$ | $0.624 \pm 0.010$ | $8.656 \pm 0.007$ |
| HK-ISVGD (OURS) | $\mathbf{4.075 \pm 0.035}$ | $4.824 \pm 0.113$ | $\mathbf{0.089 \pm 0.001}$ | $\mathbf{4.094 \pm 0.014}$ | $\mathbf{0.616 \pm 0.009}$ | $\mathbf{8.611 \pm 0.007}$ |

## G  Model Architectures and Some Experiments Settings on DGM

We provide some experimental details of image generation here. Our implementation is based on TensorFlow with a Nvidia 2080 Ti GPU.

Table 5: Average test log-likelihood ($\uparrow$) for UCI regression with parameter generator.

| | COMBINED | CONCRETE | KIN8NM | PROTEIN | WINE | YEAR |
|---|---|---|---|---|---|---|
| SVGD | $-2.832 \pm 0.009$ | $-3.064 \pm 0.034$ | $0.964 \pm 0.012$ | $-2.846 \pm 0.003$ | $-0.997 \pm 0.019$ | $-3.577 \pm 0.002$ |
| HK-SVGD (OURS) | $-2.827 \pm 0.009$ | $\mathbf{-3.015 \pm 0.037}$ | $0.976 \pm 0.007$ | $-2.838 \pm 0.004$ | $-0.958 \pm 0.021$ | $\mathbf{-3.559 \pm 0.001}$ |
| HK-ISVGD (OURS) | $\mathbf{-2.826 \pm 0.008}$ | $-3.073 \pm 0.052$ | $\mathbf{0.989 \pm 0.008}$ | $\mathbf{-2.823 \pm 0.003}$ | $\mathbf{-0.943 \pm 0.018}$ | $-3.565 \pm 0.001$ |

For CIFAR-10 and STL-10, we test them on 2 architectures: DC-GAN based [39] and ResNet based architectures [40, 41]. The DC-GAN based architecture contains a 4-layer convolutional neural network (CNN) as the generator, with a 7-layer CNN representing $h_{\boldsymbol{\phi}}^t$ in (1) and (2). In the ResNet based architecture, the generator and $h_{\boldsymbol{\phi}}^t$ are both 10-layer ResNet. For ImageNet, we use the same ResNet based architecture as CIFAR-10 and STL-10. For CelebA, the generator is a 10-layer ResNet, while $h_{\boldsymbol{\phi}}^t$ is a 4-layer CNN.

For CIFAR-10, STL-10 and ImageNet, spectral normalization is used, while we scale the weights after spectral normalization by 2 on CIFAR-10 and STL-10. We set $\beta_1 = 0.5$, $\beta_2 = 0.999$ for the Adam optimizer and $m = 1$, $n = 64$ in Algorithm 2. Only one step Adam update is implemented for solving (7). Output dimension of $h_{\boldsymbol{\phi}}^t$ is set to be 16. For all the experiments with kernel (2), both $f_{\boldsymbol{\psi}_1}^t$ and $f_{\boldsymbol{\psi}_2}^t$ are parameterized by 2-layer fully connected neural networks.

For CelebA, we scale the kernel learning objective, i.e. (7), by $\sigma$ in (11) as SMMD. Spectral regularization [26] is used. We set $\beta_1 = 0.5$, $\beta_2 = 0.9$ for the Adam optimizer and $m = 1$, $j = 5$, $n = 64$ in Algorithm 2. Only one step Adam update is implemented for solving (7). Output dimension of $h_{\boldsymbol{\phi}}^t$ is set to be 1, because scaled objective with $h_{\boldsymbol{\phi}}^t$ dimension larger than 1 is time consuming.

As for evaluation, CIFAR-10, STL-10 and ImageNet are evaluated on 100k generated images, while CelebA is evaluated on 50k generated images due to the memory limitation.

## H  More Results on Image Generation

(a) HK                    (b) HK-DK

Figure 2: Generated images on CIFAR-10 ($32 \times 32$) with DC-GAN architecture.

(a) HK            (b) HK-DK

Figure 3: Generated images on CIFAR-10 ($32 \times 32$) with ResNet architecture.

(a) HK            (b) HK-DK

Figure 4: Generated images on STL-10 ($48 \times 48$) with DC-GAN architecture.

(a) HK

(b) HK-DK

Figure 5: Generated images on STL-10 ($48 \times 48$) with ResNet architecture.

(a) HK

(b) HK-DK

Figure 6: Generated images on ImageNet ($64 \times 64$).

(a) HK

Figure 7: Generated images on CelebA $(160 \times 160)$.