[Reviews · NeurIPS 2020]

Review 1

Summary and Contributions: This work is closely related to previous works like MMD-GAN [8] and combines the idea from heat kernel, manifold learning, and Wasserstein Gradient Flows (WGFs) [7]. [8] itself is also motivated from previous work GMMN from Yujia Li, Kevin Swersky, and Richard Zemel. Generative moment matching networks, ICML 2015. Section 3 first proposed a theoretical foundation for heat-kernel learning by learning the PDF on manifold by the corresponding WGF [7]. Also a practical heat-kernel learning algorithm is proposed as algorithm 1 (and algorithm 2 with details in Appendix), and can be applied into different applications such as (1) help learn the kernel in SVGD, (2) help learn bot the generator and the kernel in Deep Generative Models. Experimental results in section 4 includes toy data, improved SVGD for BNN regression on UCI data, and image generation on 4 computer vision data sets. Also there are more theoretical analysis, algorithm and experiment details presented in Appendix. 

Strengths: Implicit manifold learning by the explicit kernel learning is an interesting and reasonable idea (i.e., there are previous work in this direction), in particular, this work apply manifold kernel learning to DGMs to high quality image generation as one quite interesting application. This work show some previous like MMD-GAN[8] can be views as an special case of the proposed framework.  The theoretical analysis on heat-kernel and HK algorithm developed based on the WGF framework [7] (section 2 & 3 & Appendix) is quite nice, and agree with the authors as "inspire researchers in deep learning to borrow ideas from other fields (like mathematics, physics, etc.)" said the broader impact section. 

Weaknesses: It seems there are closely related works are not cited and discussed in the paper (see Relation to prior work). As mentioned in section 1 "Once the heat kernel is learned, it can be directly applied to a large family of kernel-based machine learning models", it seems will be better we can see more supports for this argument for example combine the proposed HK algorithm with SVM or other kernel machines, in additional to SVGD and DGM. In particular, apply proposed HK (or HK-DK) to DGMs is a quite details involved process vs. it will be more clear to evaluate the performance of proposed heat-kernel with relative clean kernel methods together.  As one key empirical results in section 4.3 for Deep Generative Models, there are some concerns for the proposed HK & HK-DK. (1) HK-DK has convergence issues for CelebA, raise questions for the numerical stability, (2) for most of the data sets, it seems MMD-GAN with repulsive loss [28] achieved very similar performance with HK or HK-DK under IS score, and there is no failure case for [28]. Also, under the metric IS + data CIFAR-10 + RetNet, it seems Auto-GAN [34] achieved better performance than proposed HK and on bar with HK-DK. (3) Furthermore, there is no clear quantitative metric to evaluate the GAN for image generation, for example IS is proposed in [31] but seems no theoretical guarantee for metric fairness. [31] also claims it is important to get a large enough number of samples, and the 50K or 100K generated images in section 4.3 are probably at the borderline. 

Correctness: Yes

Clarity: Yes

Relation to Prior Work: This submission cited related works in several directions such as manifold learning, deep learning, GAN, etc. However, the citation is often not precise, for example one important related work is MMD-GAN [8], which is a NIPS 2017 paper but only listed as "Chun-Liang Li, Wei-Cheng Chang, Yu Cheng, Yiming Yang, and Barnabás Póczos. Mmd gan:Towards deeper understanding of moment matching network, 2017" in the reference section (page 9). Similar issues happen for [9], [27] and other references too, I feel this can be improved by the authors.  Also, there are previous work on kernel learning with the consideration to explore the manifold structure, in general these works are not cited enough and no clearly comparison with them. 2 examples (e.g.,there are many of them) are "Learning a kernel matrix for nonlinear dimensionality reduction", by K. Q. Weinberger, F. Sha, and L. K. Saul at ICML 2004, and "Kernel Learning for Extrinsic Classification of Manifold Features" by R. Vemulapalli, J. K. Pillai, R. Chellappa at CVPR 2013. It seems some of works here (may with some tweaks) can be also applied to experiments in section 4.3.

Reproducibility: Yes

Additional Feedback: Update (after rebuttal): authors did a good job to provide the feedback for (1) related works (2) experimental results as 2 major issues I mentioned. Based on this and read comments from other reviews, I keep my original rating for this paper as "6 - marginally above the acceptance threshold", and I want to see these changes author proposed in their final camera ready version to NeurIPS.


Review 2

Summary and Contributions: This paper presents a new approach to kernel learning, which leverages relations between the heat kernel and Wassterstein gradient flows to interpret the learned kernel function as approximating the heat kernel on an underlying manifold. Due to the strong relations between heat kernels and manifold geometries, this approach can be considered as a form of implicit manifold learning. The learned kernels are evaluated in two important tasks (Bayesian regression and deep generative model training), where the proposed approach outperforms previous methods.

Strengths: The method itself presented here (specifically, in section 3.2) is a kernel learning approach, building on neural network formulation from previous work in references [8,10,11]. The kernel here is derived to approximate the heat kernel on a hidden underlying manifold by considering a fundamental relation established in [19] between heat kernels on manifolds and Wasserstein gradient flows. This core relation is reiterated here in Theorem 2, showing how to relate solutions of the heat equation, given by heat kernel operator applied to dirac initial conditions, to trajectories of the Wasserstein gradient flow guided by relative entropy. This core relation is then related on one hand to manifold learning, by the rich and well studied relations between Riemannian manifold geometries and heat diffusion over them, and on the other hand to the kernel learning via parameterization and approximation of the gradient flow trajectories with neural networks. This latter part is addressed in section 3.1 that reformulates a result from [19] in terms of probability density functions that can be parameterized as neural networks, instead of abstract measures. To demonstrate the utility of their kernel learning approach, the authors present applications in the context of SVGD and deep generative models, which outperform competing methods on several tasks and datasets.

Weaknesses: Theorem 3 seems like a somewhat straightforward reformulation of Proposition 4.4 from [19], where the only difference is whether the trajectory is considered as a sequence of measures or PDFs. Importantly, this does not reflect on whether the PDFs can be parameterized by a finite neural network, or what would be the approximation error of the network approximation. More generally, following Theorem 2, both p^t and q^t are in fact instances of the heat kernel k_M^t, just applied to two different initial conditions (with a dirac positioned in different starting point). This makes lines 139-140 imprecise, and perhaps even misleading, as this theorem has nothing to do with an approximation of the heat kernel. I would recommend revising the explanation here as addressing the reformulation of the WGF trajectories in terms of pointwise functions rather than measures that are defined over sigma algebras, thus enabling the formulation of the heat kernel objective in the next section. In lines 114-115, It is unclear why the authors find negative entropy clearer than just using the original relative entropy terminology from [19].

Correctness: Generally, both the theoretical and applicative aspects of this work seem sound. The authors should clarify in section 3.1 which initial condition is considered implicitly by the notation k_M^t. Furthermore, they should clarify both p^t and q^t there (from Theorem 3) can serve as k_M^t, as they are both given by the heat kernel, just with different initial conditions.

Clarity: Generally, the paper is clear and well written.

Relation to Prior Work: The introduction addresses appropriately related and previous work, while empirical comparisons demonstrates advantages over them.

Reproducibility: Yes

Additional Feedback: *** Updates following author response *** I would like to thank the authors for the clarifications in their response, which they indicate will be added to the final version of the paper. Since I already recommended acceptance of the paper, no change is needed in the score and I hope to see it presented in the conference. === Original review: === The theoretical derivation provided here is based on a dirac initial condition, while the practical algorithm is based on a different initial condition approximating some data distribution. While the authors try to explain this as summing or aggregating together multiple trajectories, perhaps it would be useful to try and reformulate the derivation to begin with using a different initial condition matching this non-dirac distribution.


Review 3

Summary and Contributions: The paper propose to learn the heat kernel of an unknown manifold in order to capture the geometry of a data manifold. The heat kernel is parametrized by neural networks (thereby avoiding the out-of-sample issue plaguing earlier graph-based approaches) and is then used in kernel based algorithms. The paper brings together existing approaches in a novel and interesting way. Empirical results look promising. After the rebuttal: I acknowledge the rebuttal, and stand by my initial review.

Strengths: The proposed approach is mathematically natural, and a series of reasonable approximations are proposed that lead to a practical numerical algorithm. The paper does a nice job of introducing previous theoretical work, and through new theoretical developments bridging them together to form a new approach.

Weaknesses: My main concern with the paper is the assumption that the data manifold has positive curvature. This is a very strong assumption that most surely will not be satisfied for most data manifolds. This limitation should not prevent publication but it would be good to discuss more explicitly in the paper. At least, I'd like to better understand: (*) Is the assumption only needed for proving the theory or is it such that the algorithmic approach can only learn heat kernels associated for positively curved manifolds? (*) Does the assumption lead to a bias? Is the algorithm skewed towards learning certain (positively curved) manifolds due to this assumption?

Correctness: The work appears correct.

Clarity: The paper is nicely structured and easy to follow considering that it is a mathematically somewhat dense paper. I would recommend that the authors consult with a native English writer to help with grammar (as a non-native English speaker myself, I can relate to the difficulty of getting the grammar correctly).

Relation to Prior Work: To the best of my knowledge the authors did a good job of surveying the relevant literature. At ICML 2020 there was a related paper: Variational Autoencoders with Riemannian Brownian Motion Priors Dimitris Kalatzis, David Eklund, Georgios Arvanitidis and Søren Hauberg. In International Conference on Machine Learning (ICML), 2020. This considers diffusion priors on learned manifolds. It touches on many of the same ideas as the authors, but in a different setup. It would be good to acknowledge this work, but it does not limit the novelty of the current paper.

Reproducibility: Yes

Additional Feedback: I found the "Broader Impact" section to be of quite limited scope. I know that this is a new section with no prior work to lean on, but I think the authors can do better than merely state that their work will have impact.


Review 4

Summary and Contributions: This paper presents a concept named implicit manifold learning to learn the information of manifold through a heat kernel approximated by deep neural network for its great capability in encoding geometric contents in manifolds. The paper gives a theoretical analysis of the proposed method. The paper also did case studies on the application of BNN regression, deep GANs. The application model outperforms its competitor methods. However, the reviewer is not very familiar with the area discussed in this paper and is not able to judge the contribution comprehensively.

Strengths: 1) This paper is making attempts to build a general framework for manifold learning. It gives several application cases that show the effectiveness and superiority of the proposed framework. 2) This paper presents a theoretical analysis of the proposed kernel when it’s applied in MMD. It studies the continuity in weak-topology, which could ensure good convergence of the generator.

Weaknesses: 1) I appreciate the methods supported by theoretical grounds. However, considering that the effectiveness of the proposed framework is verified by empirical results, I believe that statements describing the experimental settings are still necessary. For example, in the application parts, the pseudo-code that describes how the proposed method is deployed in the existing model is desired. Currently, it’s not easy to understand how to use the proposed method to assist existing applications, like GANs. 2)The experimental results are not completely convincing. Although the results are from several different datasets, it seems that most of the images are down-scaled. For example, the images from ImageNet is down-scaled to 64*64, which are much easier than the high-resolution version of them. So I am not sure if the proposed method can also work well in high resolution cases, e.g. images from FFHQ dataset. Such modifications could also lead to different statistical results. It seems that some of the results of the compared methods are directly copied from the original paper, which means inconsistent experimental settings. I thus suspect the fairness of the comparisons. Please clarify this.

Correctness: Addressed by the author feedback --------updated---------- The claims about experimental results are not well-supported

Clarity: It takes some efforts to understand current version of the paper.

Relation to Prior Work: Yes. However, the references need to be improved.

Reproducibility: Yes

Additional Feedback: I have read the comments from other reviewers and the rebuttal message from the authors. As the authors stated, the implementation details of the application have been shown in the supplementary materials, and the authors have made sure that the experimental comparisons are fair. And as pointed out by the other reviewers, the paper has several advantages. The authors have done a good job in addressing most of my concerns. Thus, I will increase the score. -------------------------updated----------------------------------------- No previous message

[Author Response · NeurIPS 2020]

We would like to thank all the reviewers for their thoughtful and generally positive comments. We address their concerns below, and will make corresponding clarifications in the revision.

### To Reviewer 1:

**Q:** Related works.

**A:** Thanks for pointing this out. We will include more related work and cite them more precisely. We have already discussed in the paper the related kernel-based DGMs, such as MMD-GAN[1], SMMD-GAN[2], Repulsive loss [3]. We will add more kernel learning methods such as IKL [4]. Even comparing to these results, ours is still much better.

**Q:** Experimental results.

**A:** Our proposed method learns the kernel using WGF. The method itself is independent of the kernel network construction. HK and HK-DK differ only at their network constructions, and HK-DK is based on kernel approximation using Fourier features [5]. The problem of HK-DK comes from sampling; it needs much more samples to ensure a good kernel approximation in high-resolution case, which can lead to training instability. Thus, the convergence issue of HK-DK does not affect the novelty and effectiveness of our proposed method. As for evaluation, we agree that there is no theoretical guarantee on the metrics, and the metrics may not be perfect. However, there is no better metric available as far as we know. Following existing works, we evaluated all the methods on both FID and IS, under the same settings, with the same model structures and datasets. We believe that it is fair enough to show the effectiveness of our method. Although Auto-GAN achieves comparable IS, its FID score is much worse than ours (i.e., 20.4% higher on CIFAR-10 and 29.5% higher on STL-10). Furthermore, Auto-GAN needs architecture search, which costs 43 hours according to the authors, while ours needs only 12 hours to obtain better results.

### To Reviewer 2:

**Q:** Theorem, initial condition and explanation.

**A:** Thanks for the suggestions. We agree that all the PDFs can be seen as instances with different initial conditions and it is pointwise in practice. By approximating the heat kernel, we mean approximating the function with one specific initial condition (e.g. with a specific data point as the 'heat source'). We will rephrase the sentences accordingly to avoid confusion.

**Q:** Negative entropy v.s. relative entropy.

**A:** We use negative entropy because relative entropy is often referred to KL-divergence in some research domains, which may lead to confusion. We will emphasize this in the revision.

### To Reviewer 3:

**Q:** About the assumption.

**A:** The assumption is used for proving the exponential convergence. It is related to the convexity property of the functional. We agree that the manifold could be much more complicated in practice, and our proposed method is still effective for those general manifolds (as demonstrated in the experiments). The algorithm is not biased. It only means that we cannot provide guarantees on the convergence rate and approximation error without the assumption.

**Q:** Reference and broader impact.

**A:** Thanks for the suggestion. We will discuss more recent works and impact.

### To Reviewer 4:

**Q:** Experiments, pseudo-code and reproducibility.

**A:** Due to space limit, we provided pseudo-code of the deep generative model, experimental settings and hyper-parameters in the Appendix. The code is contained in the supplementary files with running scripts. We would like to emphasize that our model structures and data resolutions all follow previous works [2, 6] for fair comparisons. These works provided detailed information of their experimental settings and code online. Our settings and resolutions are consistent with theirs. Please kindly check our Appendix and code to verify this. The reported experimental results of all compared methods are under exactly the same settings. We copied some results directly from their papers whenever we cannot reproduce their results using their provided methods/codes. Thus, we believe that the comparisons are fair and our claims are conclusive.

## References

[1] Li *et al.* Mmd gan: Towards deeper understanding of moment matching network. In *NeurIPS*, 2017.

[2] Michael Arbel, Dougal Sutherland, Mikołaj Bińkowski, and Arthur Gretton. On gradient regularizers for mmd gans. In *NeurIPS*, pages 6700–6710, 2018.

[3] Wei Wang, Yuan Sun, and Saman Halgamuge. Improving mmd-gan training with repulsive loss function. In *ICLR*, 2018.

[4] Chun-Liang Li, Wei-Cheng Chang, Youssef Mroueh, Yiming Yang, and Barnabas Poczos. Implicit kernel learning. In *AISTATS*, pages 2007–2016, 2019.

[5] Yufan Zhou, Changyou Chen, and Jinhui Xu. Kernelnet: A data-dependent kernel parameterization for deep generative modeling, 2019.

[6] Takeru Miyato, Toshiki Kataoka, Masanori Koyama, and Yuichi Yoshida. Spectral normalization for generative adversarial networks. In *ICLR*, 2018.


[Meta-Review · NeurIPS 2020]

Four knowledgeable reviewers support acceptance of the paper in view of the strong theoretical analysis it provides for learning the heat kernel on a manifold via Wasserstein Gradient Flow, and its compelling performance in applications to SVGD and deep generative modeling. The paper is therefore accepted, and we ask the authors to implement the changes they proposed to make in their rebuttal before submitting the camera ready version.